



# Modelling emission and transport of key components of primary marine organic aerosol using the global aerosol-climate model ECHAM6.3–HAM2.3

Anisbel Leon-Marcos[1], Moritz Zeising[2], Manuela van Pinxteren[3], Sebastian Zeppenfeld[3], Astrid Bracher[2,4], Elena Barbaro[5], Anja Engel[6], Matteo Feltracco[7], Ina Tegen[1], and Bernd Heinold[1]

[1]Modelling of Atmospheric Processes Department, Leibniz-Institute for Tropospheric Research, 04318 Leipzig, Germany
[2]Alfred Wegener Institute, Helmholtz Centre for Polar and Marine Research, Bremerhaven, Germany
[3]Atmospheric Chemistry Department (ACD), Leibniz-Institute for Tropospheric Research, 04318 Leipzig, Germany
[4]Institute of Environmental Physics, University of Bremen, Bremen, Germany
[5]Institute of Polar Sciences - CNR, Venice, Italy
[6]GEOMAR Helmholtz Centre for Ocean Research, 24148 Kiel, Germany
[7]Department of Environmental Sciences, Informatics and Statistics, Caʹ Foscari University of Venice, Via Torino, 155, 30172, Venice Mestre, VE, Italy

**Correspondence:** Bernd Heinold (heinold@tropos.de)

**Abstract.** Primary marine organic aerosol (PMOA) contributes significantly to the aerosol loading over remote oceanic regions, where sea spray dominates aerosol production in the lower troposphere, and plays an important role in aerosol-cloud-climate interactions. The sea-atmosphere transfer of organic components depends on their abundance at the ocean surface and their physicochemical characteristics. We introduce a novel approach for representing the ocean concentration of the most abundant

organic groups in seawater that are relevant to aerosols. By apportioning the phytoplankton exuded dissolved organic carbon, modelled in the biogeochemistry model FESOM2.1-REcoM3, three biomolecule groups are computed (dissolved carboxylic acidic containing polysaccharides (PCHO), dissolved combined amino acids (DCAA), and polar lipids (PL)). The transfer of these marine groups to the atmosphere is represented by the OCEANFILMS (Organic Compounds from Ecosystems to Aerosols: Natural Films and Interfaces via Langmuir Molecular Surfactants) parameterization which is implemented in the

aerosol-climate model ECHAM6.3–HAM2.3 to represent the emission and transport processes in the atmosphere. The concentration of biomolecules in the ocean serves as the bottom boundary condition for the PMOA simulation within the aerosol model. Among the simulated organic groups in seawater, modelled PCHO is the most prevalent, followed by DCAA and PL. Conversely, PL contributes the most to the organic matter in aerosols, given the high air-seawater affinity of lipids compared to the other groups. Biomolecules exhibit minor variations in Equatorial waters, whereas strong seasonal patterns are observed to-

wards the polar regions. The global aerosol model simulations indicate that PMOA emission fluxes are primarily influenced by marine biological activity and surface wind conditions. Based on the most comprehensive evaluation to date, the computed levels of biomolecules in the ocean and species-resolved PMOA concentrations are compared with ground-based measurements across the globe. The comparison shows a strong agreement, given the uncertainties in model assumptions and measurements. Since PMOA is emitted together with sea salt, model biases in the representation of the marine organic aerosol groups are

caused by uncertainties in the simulated sea salt concentrations. A comparison with a set of long-range in-situ aircraft mea-





surements indicates that by including PMOA in the model, the representation of organic aerosols in the Southern Oceans is significantly improved.

# 1 Introduction

Oceans are a major source of natural aerosols (O'Dowd et al., 1997; Simó, 2004; Lewis and Schwartz, 2004; Galí et al., 2018;
Rinaldi et al., 2020). Wind-generated sea spray aerosol (SSA) particles predominate in the marine boundary layer (Blanchard and Woodcock, 1980; O'Dowd et al., 1997; Lewis and Schwartz, 2004). They therefore significantly influence the climate system through aerosol-radiation and aerosol-cloud interactions in remote marine and coastal regions (Pandis et al., 1994; Murphy et al., 1998; Carslaw et al., 2013; Vergara-Temprado et al., 2017).

Multiple experimental studies simulating air bubble bursting have demonstrated that marine organic constituents are co-emitted
with sea salt in sea spray (Keene et al., 2007; Facchini et al., 2008; Schmitt-Kopplin et al., 2012). This so-called primary marine organic aerosol (PMOA) emitted directly this way dominates the submicron range of the sea spray particle size distribution (Facchini et al., 2008; Gantt et al., 2011; Gantt and Meskhidze, 2013). Nonetheless, organics also contribute to the chemical composition of coarse mode of sea salt aerosol particles (Hawkins and Russell, 2010; Russell et al., 2010; Leck et al., 2013; Zeppenfeld et al., 2021).

Many organic compounds detected in ambient marine samples are found to be highly enriched in the surface microlayer (SML) with respect to bulk water (Engel et al., 2017; Pinxteren et al., 2017; Triesch et al., 2021a, b; Zeppenfeld et al., 2023). The SML is the uppermost layer of the ocean, which often contains high concentrations of organic compounds that cover the surface of raising bubbles before bursting (Stefan and Szeri, 1999; Sellegri et al., 2006; Bigg and Leck, 2008). The characterized fraction of organic matter in the ocean is dominated by lipid-like, polysaccharidic and proteinaceous compounds (Wakeham et al.,
1997; Repeta, 2015) that have also been detected inside aerosol particles (Frossard et al., 2014; van Pinxteren et al., 2023). The formation of PMOA is determined by the physicochemical properties of marine organic compounds, with the transfer from bulk water to SML to the atmosphere occurring in a chemo-selective manner (Facchini et al., 2008; Schmitt-Kopplin et al., 2012; Burrows et al., 2014). Highly surface-active molecules are preferably transferred compared to non-surface-active constituents. Thus, a differential enrichment is found in the aerosols compared to their analogous in seawater (Rastelli et al.,
2017; van Pinxteren et al., 2023).

Since a long time, there has been a high level of interest in the modelling of the emission, transport and physicochemical properties of PMOA in the context of aerosol-climate studies (O'Dowd et al., 2008; Vignati et al., 2010; Long et al., 2011; Albert et al., 2012; Gantt et al., 2012a; Vergara-Temprado et al., 2017). Various parametrizations, based on chlorophyll-$a$ (chl-$a$) ocean concentration as a proxy for marine biological activity, have been used to account for wind-driven emissions of
PMOA (O'Dowd et al., 2008; Gantt et al., 2011; Long et al., 2011; Rinaldi et al., 2013). Nevertheless, chl-$a$ does not correlate to the organic fractions in aerosol particles in some regions (Prather et al., 2013; Collins et al., 2016), especially in oligotrophic waters. Hence, a more physically based framework to parameterize the coverage of surfactants on the bubble film and the relative enrichment in the aerosols was introduced by Burrows et al. (2014). The scheme requires input data that account for



the presence of the most abundant macromolecules in the ocean. To this end, an ocean biogeochemistry model is needed to
compute these quantities (Burrows et al., 2014; Ogunro et al., 2015).

Both methods of estimating PMOA mass fractions have been included and validated in global models (Meskhidze et al., 2011; Gantt et al., 2012b; Gantt and Meskhidze, 2013; Huang et al., 2018; Han et al., 2019). Recently, Zhao et al. (2021) found that the scheme by Burrows et al. (2014) yields to an improvement in the representation of PMOA compared to a chl-$a$ based approach. Some modelling studies have additionally implemented the ability of PMOA to serve as ice nucleating particle
(INP) based on empirical formulations (Wilson et al., 2015; DeMott et al., 2016; McCluskey et al., 2018b). Special attention has been given to the significance of PMOA in the climate system as relevant INP and, to a lesser extent, as cloud condensation nuclei (CCN), especially over remote marine environments (Gantt et al., 2012b; Burrows et al., 2013; Yun and Penner, 2013; Huang et al., 2018; Vergara-Temprado et al., 2018; Zhao et al., 2021; Burrows et al., 2022). The characteristics of the chemical compounds are also assumed to be important for their ice formation potential. There is evidence of high ice activity for marine
polysaccharidic and proteinaceous compounds compared to other measured organic groups (McCluskey et al., 2018a; Alpert et al., 2022). Thus, representing PMOA as an independent component is crucial to establishing a solid foundation for future research on the climate impact of these biological compounds, particularly their effects on mixed-phase clouds.

To further investigate the occurrence of different PMOA species in various climate regions, we implement the sophisticated PMOA emission scheme by Burrows et al. (2014) into the global aerosol-climate model ECHAM6.3–HAM2.3 (Tegen et al.,
2019). With regard to the marine boundary conditions, an approach for calculating the most important organic compounds in the ocean is introduced. In this approach, the contribution to the dissolved organic carbon (DOC) from the phytoplankton exudation is considered in contrast to carbon release via cell lysis by Burrows et al. (2014). The approach is based on simulation results from the detailed marine biogeochemical model FESOM2.1-REcoM3 (Gürses et al., 2023). This open-source marine biogeochemistry model allows for regional grid refinement, which improves the spatial and temporal representation of ocean
biogeochemical processes, such as the evolution of phytoplankton blooms. This provides the basis for future high-resolution and species-resolved PMOA modelling studies to improve the model representation and understanding of marine aerosols and their interactions with different cloud types.

In this work, the so extended aerosol-climate model is thoroughly evaluated against observations worldwide, and the results are analysed globally in terms of temporal and spatial patterns. Through in-situ aircraft measurements and a more component-
specific description for ground-based observations, a comprehensive assessment of the total PMOA can be achieved.

The paper is organized as follows. Section 2 introduces the approach considered to compute the organic aerosol mass fraction and the concentration of the biomolecules in the ocean. Section 3 presents a description of the aerosol-climate model, the aerosol model setup and experiments. Section 4 describes the observational data used for the evaluation. Figure 1 shows a condensed illustration of the model components of this study for all compartments and encapsulates what is presented in
Sect. 2–4. Sections 5 and 6 discuss the model results and the comparison with measurements focusing on marine biomolecules at the sea surface and the associated aerosol emissions and transport, respectively. The computed and analysed parameters in these sections are also included in Figure 1. Finally, Sect. 7 summarizes and draws the conclusions of this work.



## 2 Modelling marine organic aerosol emissions and marine biomolecules

To represent the mass fraction in marine nascent aerosol, we base our calculations on OCEANFILMS (Organic Compounds
from Ecosystems to Aerosols: Natural Films and Interfaces via Langmuir Molecular Surfactants; Burrows et al., 2014). For
the present study, we included minor adaptations to the scheme. A detailed description of the assumptions made here together
with the calculation of the marine biomolecule groups in the ocean is given in the following sections.

### 2.1 PMOA emission parameterization

OCEANFILMS is a modelling framework that represents the sub-micron organic mass fraction in sea spray aerosol (Air-sea
interface compartment of Fig. 1). It is based on the Langmuir isotherm to represent the adsorption at bubble surfaces of the
marine organic matter, which is apportioned into several classes. They include lipid-, polysaccharide-, protein-, humics-, and
processed-like mixtures. The last two, humics- and processed-like mixtures, describe the recalcitrant DOC at the ocean surface,
which is the DOC fraction that accumulates due to its resistance to rapid bacterial degradation (Hansell et al., 2012). These
classes possess differing physicochemical characteristics: molar weight ($MW_i$), surface area ($A_i$), carbon concentration in
seawater ($C_i$) and Langmuir adsorption ($\alpha_i$). $C_i$ and $\alpha_i$ regulate the bubble fractional surface coverage ($\theta_i$). Note that $i$ stands
for the different classes. In combination with the bubble coating parameter (n = 2, equivalent coverage of the interior and
exterior of the bubble), the mass on the bubble surfaces ($M_i$) for each class can be computed as:

$$M_i = n\theta_i \frac{MW_i}{A_i} \tag{1}$$

Since PMOA and sea salt (SS) are emitted together, they make the total mass of sea spray aerosol (SSA) ($M_{SSA}$). The organic
mass fraction ($OMF_i$) is then calculated based on the mass of individual macromolecule group ($M_i$) and the mass of sea salt
($M_{SS}$) per bubble surface area:

$$M_{SSA} = M_i + M_{SS}, \tag{2}$$

$$OMF_i = \frac{M_i}{M_i + M_{SS}}, \tag{3}$$

where $M_{SS}$ is assumed to be constant with a value of $3.59 \times 10^{-3}\ \mathrm{g\,m^{-2}}$.

The organic carbon aerosol enrichment is a result of the differing properties of the macromolecules in seawater that regulate
their transfer to the atmosphere (Burrows et al., 2014). Despite lipids having the lowest concentration in the ocean, their
surface affinity and competitive adsorption favour their presence at the air-water interface (Frka et al., 2012) leading to higher
aerosol enrichment compared to other macromolecules. Polysaccharides and proteins surface affinity, on the other hand, is





lower, limiting their transfer to the aerosols, with enrichment factors of two orders of magnitude smaller than that for lipids (van Pinxteren et al., 2023). Lastly, humic- and processed-like mixtures have very low surface affinity and, compared to the other classes, their contribution to the marine organic aerosol mass fraction may be negligible (Burrows et al., 2014). Hence, neglecting the recalcitrant portion of DOC will not impact the OMF estimations and is therefore not considered in this study.

In the current study we account for the presence of three main biomolecule groups that represent a portion of the lipid-,

protein- and polysaccharide-like classes. We focus on the contribution of extracellular DOC from phytoplankton, apportioning the organic matter into the most abundant biomolecule groups in seawater based on a closure approach that will be introduced in Sect. 2.3 (Seawater compartment in Fig. 1). In this respect, our approach differs from Burrows et al. (2014), who considered the DOC primarily generated via cell lysis to compute the ocean concentration of the aforementioned groups. All parameters used for the computation of the main biomolecule groups in this work, however, are identical to those used in Burrows et al.

(2014). They describe operational laboratory compounds selected to represent the ocean macromolecules and are shown in Table 1. These parameters could be refined in future studies to better characterize the biomolecule groups presented in this study.

In the following sections, we introduce the different components and considerations to compute each marine biomolecule group's ocean concentration. Firstly, we describe the ocean biogeochemistry model selected, which represents the DOC in the

ocean. Later, we explain our closure approach to compute the marine organic groups based on the model tracers.

**Table 1.** Physicochemical parameters of the three ocean macromolecules considered in OCEANFILMS from Burrows et al. 2014

| Species | Molecular weight ($MW_i$ in $\mathrm{g\,mol^{-1}}$) | Mass per area at surface saturation ($\frac{MW_i}{A_i}$ in $\mathrm{g\,m^2}$) | Langmuir adsorption parameter ($\alpha_i$ in $\mathrm{m^3\,mol^{-1}}$) |
|---|---|---|---|
| Polysaccharides | 250,000 | 0.1375 | 90.58 |
| Proteins | 66,463 | 0.00219 | 25,175 |
| Lipids | 284 | 0.00259 | 15,205 |

## 2.2 Marine biogeochemistry model

The upper ocean biochemistry was simulated by the Regulated Ecosystem Model (REcoM3) coupled to the general circulation and sea-ice Finite VolumE Sea-ice Ocean Model (FESOM2.1). FESOM2.1 is an unstructured-mesh ocean circulation model with high spatial resolution in dynamically active regions, while including the remainder of the global ocean at a coarse reso-

lution (Wang et al., 2014; Danilov et al., 2017; Koldunov et al., 2019). REcoM3 describes the ocean biogeochemistry in terms of the physical and biological carbon cycle with two phytoplankton and two zooplankton functional types, nutrients, dissolved as well as particulate organic matter, and detritus. The phytoplankton metabolic processes are regulated via non-linear limiter functions based on the variable, intracellular nitrogen to carbon ratio (N:C ratio) following Geider et al. (1998) and modified for REcoM3 in Schourup-Kristensen et al. (2014). These functions regulate the nitrogen uptake and carbon exudation according





to the N:C ratio (c.f. Sect. A3.6 in Gürses et al. (2023); Sect. A6.1 in Schourup-Kristensen et al. (2014)).

Phytoplankton carbon is considered to partly exude organic carbon as dissolved carboxylic acidic containing polysaccharides (PCHO) alongside other dissolved organic carbon molecules Engel et al. (2020); Arnosti et al. (2021). PCHO and their aggregation product Transparent Exopolymer Particles (TEP) were included into REcoM version 1 by Schartau et al. (2007) and re-introduced for REcoM version 3 in the simulation used here, based on a mesocosm experiment by Engel et al. (2004), where

the aggregation parameter choice itself is constrained by a mesocosm experiment of diatoms (Engel et al., 2002). A parameter optimization was successfully conducted by Schartau et al. (2007) and validated with a second mesocosm experiment of coccolithophores (Engel et al., 2004). Additionally, the parameter values fit to observational studies (Table 1 in Engel et al., 2004). Additionally, this configuration is being assessed for the Arctic Ocean in a concurrent investigation. A detailed description and assessment of the REcoM version 3 performance on the global scale is available in Gürses et al. (2023).

The FESOM2.1-REcoM3 simulation was conducted for the period 1958-2019 on the so-called fARC mesh (https://gitlab.awi. de/fesom/farc) with 4.5 km resolution in the Arctic Ocean, north of $60°N$ (Wekerle et al., 2017; Wang et al., 2018; Schourup-Kristensen et al., 2018). The global mesh resolution gradually decreases from the poles towards the equator, having the subtropical waters the coarsest resolution of about 120 km (Schourup-Kristensen et al., 2018). Over the Equator and Southern Ocean, the resolution is relatively higher, between 30–40 km.

The simulation was forced with the atmospheric reanalysis data sets of JRA55-do v.1.4.0 (Tsujino et al., 2018) and initialized from temperature and salinity fields of the Polar Science Centre Hydrographic Climatology (Steele et al., 2001), as well as from initial fields of dissolved inorganic nitrogen and dissolved silicic acid concentration from the World Ocean Atlas climatology (Garcia et al., 2019a, b), and Dissolved Inorganic Carbon as well as total alkalinity from the Global Ocean Data Analysis Project (GLODAP) version 2 (Lauvset et al., 2016). Monthly output was retrieved for the period of 1990-2019, the preceding

years are considered as spin-up for the biological processes.

To obtain surface fields, we initially interpolated FESOM2.1-REcoM3 results from the original unstructured mesh to a regular grid of approximately 30 km ($0.25°$) horizontal resolution and calculated a volume-weighted mean for each grid cell over the upper 30 m of the water column. Finally, the biogeochemical model output and derived biomolecules in the ocean were interpolated to the ECHAM6.3–HAM2.3 grid and used as the bottom boundary condition of the aerosol model.


## 2.3 Organic biomolecules in seawater

The main sources of dissolved organic matter in seawater are phytoplankton exudates, carbon release via cell lysis, zooplankton grazing on phytoplankton and zooplankton excretion (Carlson, 2002). Additionally, DOC also significantly forms from particulate organic carbon biological degradation (Repeta, 2015). From these sources, phytoplankton carbon exudation is considered

a significant part of phytoplankton primary production (Myklestad, 2000). The most abundant components measured in extracellular carbon released by phytoplankton are carbohydrates (mono-, oligo- and polysaccharides), proteinogenic compounds (amino acids, proteins, and peptides), lipids (fatty acids and polar lipids such as phosphoglycerides and glycosylglycerides) and, to a lesser degree, organic acids (Lancelot, 1984; Yongmanitchai and Ward, 1993; Harwood and Guschina, 2009). Among





**Table 2.** List of abbreviations of the most relevant aerosol and seawater compounds considered in the present study, as well as, the locations of the campaign sites and observational stations used for the model evaluation.

| General terms | |
|---|---|
| PCHO | Dissolved carboxylic acidic containing polysaccharides |
| DCAA | Dissolved combined amino acids |
| PL | Polar lipids |
| **Seawater** | |
| DOC | Dissolved organic carbon |
| $DOC_{phy\_ex}$ | DOC fraction exuded by phytoplankton |
| $PCHO_{sw}$ | PCHO in seawater |
| $DCAA_{sw}$ | DCAA in seawater |
| $PL_{sw}$ | PL in seawater |
| $DCCHO_{sw}$ | Dissolved combined carbohydrates |
| $PG_{sw}$ | Dissolved phosphatidylglycerol |
| TEP | Transparent exopolymer particles |
| **Aerosols** | |
| PMOA | Primary marine organic aerosol |
| OA | Organic aerosol |
| OC | Organic carbon |
| $OM_{aer}$ | Organic mass in aerosol |
| SSA | Sea spray aerosol |
| SS | Sea salt |
| $PCHO_{aer}$ | PCHO in aerosol particles |
| $DCAA_{aer}$ | DCAA in aerosol particles |
| $PL_{aer}$ | PL in aerosol particles |
| $CCHO_{aer}$ | Combined carbohydrates |
| $CAA_{aer}$ | Combined amino acids |
| $PG_{aer}$ | Phosphatidylglycerol |



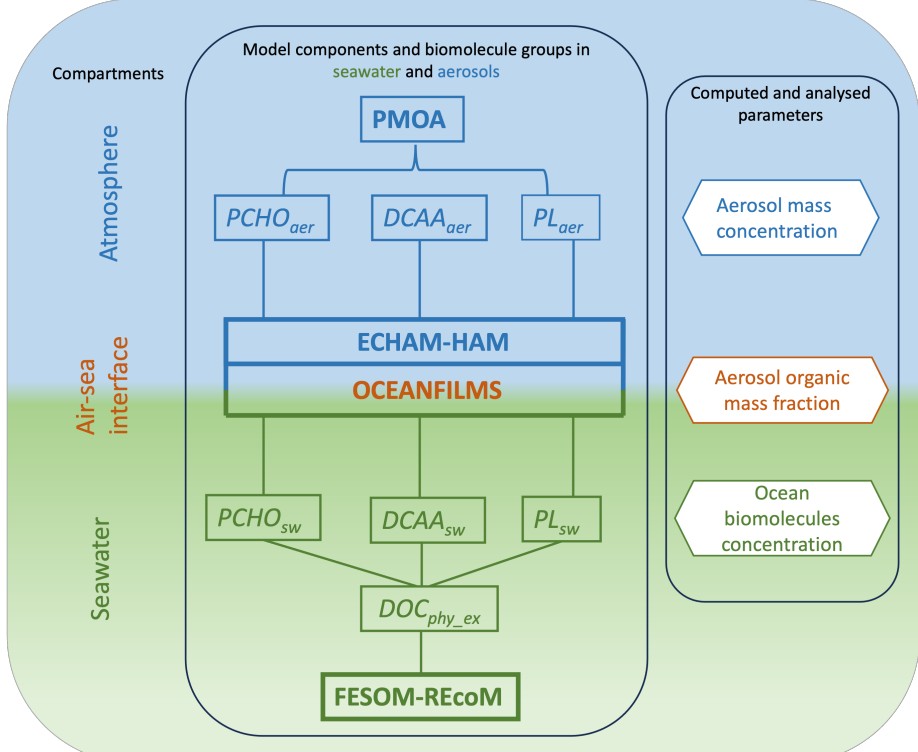

**Figure 1.** Schematic of the main modelling components considered in this study for simulating the biomolecules in seawater and their transfer to the atmosphere. Note that FESOM2.1-REcoM3 model data are used in offline mode and the modelled biomolecule groups serve as bottom boundary conditions of the integrated component OCEANFILMS+ECHAM6.3–HAM2.3. See Table 2 for the compound abbreviations.

these, polysaccharides, free and combined amino acids and polar lipids represent the main biomolecule groups in the phy-
toplankton extracellular products (Parrish and Wangersky, 1987; Parrish et al., 1993, 1994; Obernosterer and Herndl, 1995; Engel et al., 2004; Arnosti et al., 2021).

These biomolecules can be measured both in seawater and in the aerosol phase (Kuznetsova et al., 2004; Zeppenfeld et al., 2020; Triesch et al., 2021b; van Pinxteren et al., 2023). In the present study we assume the aforementioned groups to be char-acterized by the biomolecules that form the majority of the phytoplankton extracellular carbon: dissolved carboxylic acidic
containing polysaccharides (PCHO$_{sw}$), dissolved combined amino acids (DCAA$_{sw}$), and phospholipids and glycolipids as po-lar lipids (PL$_{sw}$) in seawater. The DOC fraction exuded by phytoplankton (DOC$_{phy\_ex}$) is resolved in the FESOM2.1-REcoM3 model Gürses et al. (2023), and we use it to derive the biomolecule groups. Based on these premises, we compute the ocean surface concentration of the biomolecules by apportioning the DOC$_{phy\_ex}$, into the contribution from each group:

$$DOC_{phy\_ex} = C_{PCHO_{sw}} + C_{PL_{sw}} + C_{DCAA_{sw}} + Res \tag{4}$$





$C_{PCHO_{sw}}$, $C_{PL_{sw}}$ and $C_{DCAA_{sw}}$ refer to the surface ocean concentration of each group (seawater compartment in Fig. 1), while Res is the residual that will be attributed to compounds not included in the three main classes with contribution ranging between 9 and 38 % of $DOC_{phy\_ex}$ (Hellebust, 1965; Al-Hasan and Coughlan, 1976).

    In the FESOM2.1-REcoM3 model, the DOC phytoplankton excretion rate ($DOC_{phy\_ex\_rate}$) describes the phytoplankton release per unit of time and is considered a source term in the semi-labile DOC (see Eqs. A42 and A55 in Gürses et al., 2023):

$$DOC_{phy\_ex\_rate} = (\epsilon^{C}_{phy} f_{lim,phy} PhyC_{phy} + \epsilon^{C}_{dia} f_{lim,dia} PhyC_{dia}) \tag{5}$$

    PhyC refers to the phytoplankton carbon concentration, and the sub-indices "$phy$" and "$dia$" refer to small and diatom phytoplankton groups, respectively. $\epsilon$ is the excretion constant of organic carbon $[d^{-1}]$ and f is a limiter function that downregulates the phytoplankton excretion when the nitrogen quota ($qN : C$) becomes too high.

    To represent acidic dissolved polysaccharides in seawater ($PCHO_{sw}$) and transparent exopolymer particles (TEP), which are

gel-like particles formed from $PCHO_{sw}$, Schartau et al. (2007) developed a formulation based on the extracellular production of organic carbon from phytoplankton and aggregation processes (see previous section). Simulated $PCHO_{sw}$ accounts for approximately 63 % of exuded organic carbon by small phytoplankton and diatoms, representing the majority of the modelled DOC. According to laboratory studies, dissolved polysaccharides account for the highest fraction of exuded carbon by phytoplankton, and their contribution to the exuded carbon ranges from 47 to 90 % (Myklestad, 1995; Biersmith and Benner, 1998;

Hama and Yanagi, 2001). It is therefore assumed that:

$$C_{PCHO_{sw}} = PCHO_{sw} \Big|_{FESOM-REcoM} \tag{6}$$

    Overall, lipid material in phytoplankton exuded DOC ranges from 2.8 to 10.3 % (Hellebust, 1965; Billmire and Aaronson, 1976). Considering that on average $\delta = 5$ % of the $DOC_{phy\_ex\_rate}$ will be attributed to the extracellular $PL_{sw}$ production rate ($S_{PL_{sw}}$), the ocean surface concentration can be approximated as $S_{PL_{sw}}$ multiplied by the lifetime ($\tau$) of lipids in seawater after

released:

$$C_{PL_{sw}} = \tau S_{PL_{sw}} \tag{7}$$

    with

$$S_{PL_{sw}} = \delta DOC_{phy\_ex\_rate}. \tag{8}$$

    Lipids are short-lived compounds, whose turnover time is just a few days (Hopkinson et al., 2002; Karl and Björkman,

2015). Sensitivity studies, performed with typical turnover times of $PL_{sw}$ between four and ten days ((Karl and Björkman, 2015)), led to the best agreement with observation for turnover rates of eight days.

    $C_{DCAA_{sw}}$, on the other hand, was determined differently. Based on various concentration values measured in ambient seawater





from different sites (see also the measurement description below), we calculated the ratio of observed dissolved combined carbohydrates (DCCHO$_{sw}$) and DCAA$_{sw}$ (from 31 seawater samples). A median value of $Ratio = 0.3\pm0.08$ was obtained

and is used here to compute DCAA$_{sw}$ in the ocean based on PCHO$_{sw}$ modelled concentration. Nevertheless, employing this method results in the estimated DCAA$_{sw}$ encompassing both extracellular and intracellular carbon derived from phytoplankton. Consequently, the modelled concentrations reflect the aggregate of these two DCAA$_{sw}$ formation mechanisms, as it is not feasible to differentiate the relative contribution of extracellular carbon released by phytoplankton based on observational data. Hence, in our approach, DCAA$_{sw}$ will be the sole group for which the sources may include contributions beyond extracellular

release by phytoplankton (C$'_{\text{DCAA}_{sw}}$).

$$\text{C}'_{\text{DCAA}_{sw}} = \text{Ratio} * \text{C}_{\text{PCHO}_{sw}} \tag{9}$$

Considering that carbohydrates constitute a significant portion of semi-labile DOC, with turnover times ranging from months to years, the computed DCAA$_{sw}$ will also contribute to this fraction. Therefore, the labile or refractory component of DCAA$_{sw}$ is not included in the current study.

Values found in literature indicate that extracellular amino acids represent 1.5 % to 7 % of exuded DOC (Myklestad et al., 1972; Mague et al., 1980; Granum et al., 2002). Presuming that extracellular DCAA$_{sw}$ represent nearly 5 %, together with PCHO$_{sw}$ and PL$_{sw}$, the biomolecules constitute approximately 73 % of exuded organic carbon by phytoplankton groups in FESOM2.1-REcoM3, where dissolved acidic polysaccharides account for the highest fraction. The residual 27 % may include other lipid-, polysaccharide- and protein-like compounds, as well as organic acids or other unknown components.

## 2.4 Approximations of phytoplankton extracellular carbon release

The abundance of the biomolecules exuded by phytoplankton exhibits a significant temporal and spatial variability, primarily influenced by phytoplankton growth phase and nutrient availability (Myklestad, 2000). Other studies have demonstrated that the carbon exudation also differs among species (Hellebust, 1965; Wolter, 1982; Wetz and Wheeler, 2007). Furthermore, intense light conditions induce abrupt modifications in the extracellular release, with the proportion of carbon incorporated into cells

remaining approximately constant in comparison to that exuded by phytoplankton, as documented by Mague et al. (1980).

During phytoplankton growth, extracellular carbon release is influenced by nutrient conditions. The exudation tends to be slightly higher for the rapidly growing than for the stationary phase (Myklestad et al., 1989). The exuded products differ for every case and phytoplankton species. For instance, higher levels of extracellular polysaccharides and free amino acid release were observed during phosphorus limited conditions compared to balanced nutrient conditions (Obernosterer and Herndl,

1995). In contrast, whereas proteinogenic compounds and free amino acids decrease under nitrogen depletion (Granum et al., 2002), extracellular polysaccharides are significantly favoured (Myklestad, 1995).

Biogeochemical models often parameterize the phytoplankton carbon exudation by setting a constant phytoplankton biomass loss per day (Thornton, 2014). This fraction is set to 10 % for both phytoplankton groups in FESOM2.1-REcoM3 (Gürses et al., 2023). Moreover, the exuded carbon is regulated by a limiting factor as a measure of nutrient availability, which depends





entirely on the carbon and nitrogen quota and, lastly, it is independent of light conditions (see Eq. (5)).

Furthermore, since simplifications are required for the global biogeochemistry model, diatoms and small phytoplankton do not distinguish the species within those groups. Therefore, the distinct characteristics of each phytoplankton culture, which exhibit an unequal distribution in seawater and extracellular carbon release levels (Granum et al., 2002), cannot be captured. Therefore, the values utilized in the present study to estimate the contribution of each biomolecule to the extracellular DOC released by

phytoplankton in the ocean were either averaged across multiple laboratory studies or approximated, thus limiting them to known measured quantities in the literature (Hellebust, 1965; Billmire and Aaronson, 1976; Mague et al., 1980; Myklestad, 1995; Biersmith and Benner, 1998; Hama and Yanagi, 2001; Granum et al., 2002).

## 3 Global aerosol-climate simulations

### 3.1 The ECHAM6.3–HAM2.3 model

Aerosol-climate models represent aerosol emission, transport, wet and dry deposition as well as direct and indirect radiative effects in the earth system. In this study, we used the model ECHAM6.3–HAM2.3 (Tegen et al., 2019), which combines the atmospheric general circulation model ECHAM6.3 (Stevens et al., 2013) with a spectral transform dynamical core (Lin and Rood, 1996) and the Hamburg Aerosol Module (HAM2.3) (Stier et al., 2005).

The aerosol microphysics module, HAM, is based on the M7 aerosol model (Vignati et al., 2004; Stier et al., 2005). The

aerosol species considered in the model are Sulfate ($SO_4$), organic carbon (OC), black carbon (BC), mineral dust (DU) and sea salt (SS). For OC, BC and $SO_4$, emissions are initially prescribed from anthropogenic sources and biomass burning emission inventories, and from volcanic eruptions. Wind driven DU and SS emission fluxes from dessert and the ocean, respectively, are calculated online in the model. Additionally, dimethyl sulfide (DMS) is emitted from the marine biosphere.

Aerosol species are divided into two groups of soluble and insoluble aerosol particles for a total of seven log-normal classes

according to a predefined 4-group aerosol size spectrum (Table 3). The aerosol mass and number concentration is prognosticated for each mode is also defined in Table 3. The log-normal distribution depends on the aerosol number, number median radius and standard deviation. Modes exist as soluble or insoluble. All species in a soluble mode are considered to be internally mixed, meaning that every particle is actually a mixture of all the species within the mode.

HAM includes aerosol transformation processes such as nucleation of sulfuric acid–water droplets, coagulation and con-

densation of sulfuric acid, and water uptake. In addition, deposition, aerosol interactions with clouds and radiation are also accounted for. The updated version of HAM2.3 encompasses several improvements to the aerosol processes and emission (Tegen et al., 2019) as well as to aerosol-cloud interactions (Lohmann and Neubauer, 2018).

The two-moment cloud microphysics scheme in ECHAM6.3–HAM2.3, follows Lohmann et al. (2007) and Lohmann and Hoose (2009). It allows for in-cloud and below cloud scavenging aerosol processes for liquid, ice and mixed-phase clouds.

The cloud droplet activation is based on the Köhler-theory by Abdul-Razzak and Ghan (2000). ECHAM6.3 has implemented a rapid radiative transfer model (PSrad/RRTMG) to represent the radiative interaction with aerosols and clouds (Pincus and Stevens, 2013).



**Table 3.** Aerosol modes and compounds in HAM. r denotes the radius of the respective particle size range and $\sigma$ the standard deviation.

| Size mode/Size range ($\mu m$) | Soluble/Internally mixed | Insoluble/Externally mixed |
|---|---|---|
| Nucleation* ($r \leq 0.005$) | $SO_4$ | |
| Aitken* ($0.005 < r \leq 0.05$) | $SO_4$, OC, BC | OC, BC |
| Accumulation* ($0.05 < r \leq 0.5$) | $SO_4$, OC, BC, DU, SS, PMOA | DU |
| Coarse** ($r > 0.5$) | $SO_4$, OC, BC, DU, SS, PMOA | DU |

*$\sigma = 1.59$; **$\sigma = 2.0$

PMOA is included in the model as new individual species ($PCHO_{aer}$, $DCAA_{aer}$ and $PL_{aer}$ of the Atmosphere compartment in Fig. 1) to the soluble accumulation and coarse modes. The model does not represent sea spray emission for the Aiken mode. Therefore, PMOA is initially emitted solely into the accumulation mode. Then, the particles grow by coagulation or condensation, increasing the mean geometric radii, and eventually transitioning to a larger mode.

### 3.2 Emissions of sea spray aerosol

Marine aerosols emission flux is calculated based on Eq. (2). where PMOA and SS make the total sea spray mass. Thus, the emitted PMOA mass flux of each biomolecule group (i) can be computed as:

$$\mathrm{PMOA}_{massflux}(i) = \frac{\mathrm{SS}_{massflux} * \mathrm{OMF}_i}{1 - \mathrm{OMF}_i}, \tag{10}$$

where, $\mathrm{SS}_{massflux}$ is the mass flux of sea salt emitted in ECHAM6.3–HAM2.3 following (Long et al., 2011) with sea surface temperature correction according to Sofiev et al. (2011) and, OMF referring to the organic mass fraction parameterized based on OCEANFILMS (Burrows et al., 2014), as previously described.

Following Burrows et al. (2022), we consider that PMOA is internally mixed and the number and fluxes added onto sea salt. The authors performed sensitivity studies with various combinations of the mixing state of PMOA with sea salt in the Energy Exascale Earth System Model (E3SM). They concluded that the configuration selected here led to a better agreement in the seasonal representation of organic mass compared with observations.

### 3.3 Experimental setup

The aerosol-climate model simulations are performed at T63 (approx. $1.875 \times 1.875$) horizontal resolution with a total of 47 vertical levels, which resolves the atmosphere from the surface up to 0.01 $hPa$. The model is run in nudged mode with the





Re-Analysis data from the European Centre for Medium-Range Weather Forecasts (ECMWF) known as ERA-Interim. Sea ice concentration (SIC, as the percentage of area covered by ice) and sea surface temperature (SST) monthly mean values from the Atmospheric Model Intercomparison Project (AMIP) (Taylor et al., 2000) are used as boundary conditions for the model experiments. The simulations cover a period of ten years (2009-2019) with a spin-up time of four months and output frequency
of 12 hours.

Two experiments, without and with simulated PMOA as a tracer in the model, were performed, hereafter referred to as SP-MOAoff and SPMOAon respectively. The latter, uses the biomolecule ocean surface concentration as bottom boundary conditions. For this experiment, a SIC and SST mask was applied within the sea salt emission scheme as an adjustment to be consistent with the biogeochemical model prognosticated sea ice, intending to avoid ambiguities. The mask controls when
to use FESOM2.1-REcoM3 model SIC and SST values over AMIP data. Whenever ice free (SIC $<10\%$) conditions for the marine biogeochemistry model are satisfied, the AMIP SIC values are updated during runtime to $0\%$ and SST is replaced by that from FESOM2.1-REcoM3. Note that the mask only applies when the sea salt emission scheme is called, not affecting the rest of the globe.

## 4   Observations for model evaluation

In this section, we will discuss the measurement data selected for the model evaluation. We present the seawater sample data of measured marine compounds that are being compared with the concentration of marine biomolecules in the ocean. Likewise, we validate the offline computed organic mass fraction and simulated aerosol concentration of each group with analogous compounds from observations. This species-wise evaluation will assess how well the models can represent the different biomolecules in seawater and the atmosphere. Nonetheless, the data are primarily accessible for specific locations
and do not provide an overview of marine organics' abundance in remote oceanic areas. Thus, as the final dataset for model evaluation, in-situ airborne organic aerosol concentration measurements with extensive coverage of most oceanic regions is used to provide a more thorough evaluation of PMOA.

### 4.1   Seawater samples and in-situ ground-based measurements

Modelled estimates of ocean surface concentration of biomolecule and aerosol OMF and concentration (Fig. 1) are compared to
bulk seawater samples and aerosol observations from various stations worldwide (Fig. 2). Table 5 summarizes the most relevant information of these marine and aerosol measurements. The comprehensive collection of observational data in this study was compiled considering similar sampling techniques and laboratory instruments to detect and measure the concentration of the organic compounds. Details on the compounds selected for the model evaluation are introduced in this section. Additionally, a brief description of the interpolation of model results for the comparison is presented.
Seawater samples were collected between 10 cm to 3.5 m depth, often with a plastic or glass bottle to collect the water at a specific depth. For simplicity, only measurements from the open ocean without sea-ice were included. Whereas the model mostly represents the biomolecule production from phytoplankton, the in-situ measurements do not allow to trace back the



production mechanism of these groups. Hence, modelled quantities may represent a portion of the measured biomolecules. Although the measurements are not strictly comparable to the model results, they are good indicators to validate the modelled

quantities.

We therefore selected analogous components for the model evaluation. Seawater measurements of dissolved combined carbohydrates (DCCHO$_{sw}$), dissolved combined amino acids (DCAA$_{sw}$) and dissolved phosphatidylglycerol (PG$_{sw}$) were chosen for comparison with modelled PCHO$_{sw}$, DCAA$_{sw}$ and PL$_{sw}$ respectively. DCAA$_{sw}$ is considered to be approximately equal to the measured hydrolysable dissolved combined amino acids (DHAA). On the other hand, the selection of PG$_{sw}$, was based

on the fact that phytoplankton extracellular lipids are essentially formed by phosphoglycerides and glycosyl glycerides (Yongmanitchai and Ward, 1993; Guschina and Harwood, 2009). Moreover, the presence of this compound in seawater is often correlated to phytoplankton (Triesch et al., 2021b).

Additionally, aerosol data were carefully cleaned, including only those for which a correlation with marine biological activity has been reported. Aerosol samples were collected in filters exposed at heights between 4 and 50 m.a.m.s.l. For the aerosols,

we selected the same tracers that are linked to the marine amounts. Observations of combined carbohydrates (CCHO$_{aer}$), amino acids (CAA$_{aer}$) and PG$_{aer}$ are available for comparison to the simulated aerosol concentration and organic fraction of PCHO$_{aer}$, DCAA$_{aer}$ and PL$_{aer}$ respectively (Table 5).

OMF for each group is available from OCEANFILMS while the total modelled OMF is calculated by totalling the individual biomolecule OMF. On the other hand, for the measurements, we derived OMF based on the observed marine aerosols and

sea salt mass, which was estimated as a constant rate (1/0.3061) of sodium (Na$^+$) concentration (Seinfeld and Pandis, 2006). Lastly, total OMF from observations is based on the concentration of organic mass (OM) in aerosol (OM$_{aer}$) to also capture other aerosol contributions besides marine organics. OM$_{aer}$ is derived from the measured OC, considering a ratio OM:OC=2 of remote region aerosols following Turpin and Lim (2001). Nonetheless, this ratio may vary as aerosol particles age in the atmosphere and depending on the content of water-soluble organic material (Sciare et al., 2005; Facchini et al., 2008).

For the model evaluation, simulated results are interpolated to the observation sites using a cubic triangular based interpolator, a suitable method to detect and account for gradients in the data. Note that we use monthly ocean values, which do not capture the spatial and temporal variability in marine biogeochemistry within a month (e.g. CVAO seawater samples in Triesch et al., 2021b). This can affect cases where the quantity of samples is limited and restricted to a single location, such as CVAO, where the interpolated ocean concentrations remain nearly identical. Nevertheless, a comprehensive examination of the daily

modelled PCHO$_{sw}$ against observations (not shown) for the summer of 2017, revealed an overall lower agreement with water samples than when utilizing monthly mean values.

For the aerosol comparison, we interpolated the near-surface model vertical level aerosol concentration to the coordinates of the stations. Most stations are land-based, except for NAO and WAP, in which aerosols were sampled on board of a ship. For these cases, we interpolated the simulated values for the ship trajectories and averaged over a starting and ending point in

accordance with observations. For Svalbard, filters were exposed for at least six days. Hence, interpolated model values were averaged over these period for the comparison with observations.



**Table 4.** List of abbreviations and coordinates of the locations of the measurement campaigns and observational stations used for the model evaluation.

| Abbreviation | Meaning | Coordinates |
|---|---|---|
| WAP | Western Antarctic Peninsula | $64°S - 68°S,\ 60°W - 69°W$ |
| NAO | North Atlantic Ocean | $64°N - 79°N,\ 2°E - 8°E$ |
| CVAO | Cape Verde Atmospheric Observatory | $16.9°N,\ 24.9°W$ |
| SB | Stony Brook Harbor | $40.9°N,\ 73.2°W$ |
| WMED | Western Mediterranean | $37.7°N - 37.8°N,\ 1.9°E - 2.2°E$ |
| AS | Adriatic Sea | $44.9°N - 45.1°N,\ 12.8°E - 13.3°E$ |
| PUR12 | Peruvian Upwelling Region 2012 campaign | $5°S - 16°S,\ 75°W - 82°W$ |
| PUR17 | Peruvian Upwelling Region 2017 campaign | $12°S - 75°S,\ 78°W - 69°W$ |
| NWAO | North Western Atlantic Ocean | $38°N - 41°N,\ 69°W - 73°W$ |
| SATL | Subtropical Atlantic | $20°N - 25°N,\ 19°W - 31°W$ |
| SVD | Svalbard | $78.9°N,\ 11.9°E$ |

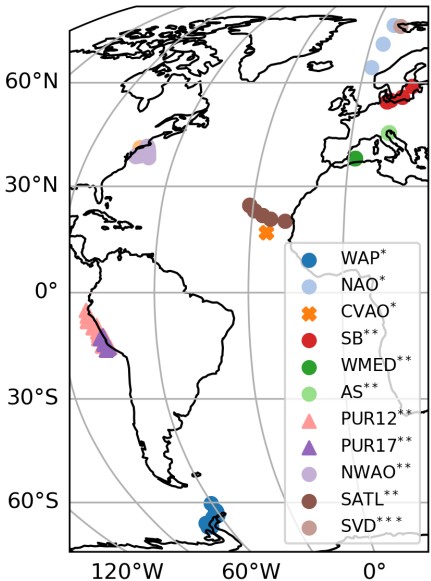

**Figure 2.** Station locations for seawater samples and marine aerosol measurements. The circles, triangles, and cross markers indicate the stations for which one, two and three compounds were measured respectively. The asterisks indicate the type of data available at each location: *, seawater and aerosol; **, only seawater; *** only aerosol. See Table 4 for the location abbreviations. The most relevant information regarding the data can be found in Table 5.





**Table 5.** Bulk water samples and aerosol measurements selected for the comparison with modelled biomolecule and aerosol tracers. Seawater sampling techniques employed during each campaign of each molecule group are the same. The measurement techniques are High-performance anion exchange chromatography coupled with pulsed amperometric detection (HPAEC-PAD) for quantifying $DCCHO_{sw}$. High-performance liquid chromatography (HPLC) for $DCAA_{sw}$, and thin layer chromatography with the flame ionization detection (TLC/FID) for $PG_{sw}$. See Table 2 and Table 4 for the compound and location abbreviations, respectively. References abbreviations: EG16- Engel and Galgani (2016); vP23- van Pinxteren et al. (2023); Z23- Zeppenfeld et al. (2023); Z21- Zeppenfeld et al. (2021); ME21- Maßmig and Engel (2021); K02- Kuznetsova and Lee (2002); K04- Kuznetsova et al. (2004); R08- Reinthaler et al. (2008); F19- Feltracco et al. (2019); T21b- Triesch et al. (2021b); F11- Frka et al. (2011).

| Compounds (seawater/aerosol) | Water samples/Aerosol availability | Location | Depth (m)/ Height (m a.m.s.l.) | No. of observations | Date | References |
|---|---|---|---|---|---|---|
| $DCCHO_{sw}$/$CCHO_{aer}$ | avail./NA. | PUR12 | 0.2 / — | 39 / — | Dec 2012 | EG16 |
| | avail./NA. | PUR17 | 1-3.5 / — | 13 / — | Apr, Jun 2017 | ME21 |
| | avail./avail. | CVAO | 1 / 30 | 3 / 7 | Sep, Oct 2017 | vP23 |
| | avail./avail. | NAO | 1 / 25 | 3 / 3 | May 2017 | Z23 |
| | avail./avail. | WAP | 0.25 / 4, 18 | 18 / 16 | Jan–Mar 2019 | Z21 |
| $DCAA_{sw}$/$CAA_{aer}$ | avail./NA. | SB | 0.1 / — | 41 / — | Feb–Sep 1999; Feb–Sep 2000 | K02 |
| | avail./NA. | NWAO | 0.15 / — | 22 / — | Jun 2001 | K04 |
| | avail./NA. | SATL | 0.3 / — | 17 / — | Sep, Oct 2004 | R08 |
| | avail./NA. | WMED | 0.3 / — | 5 / — | Sep, Oct 2003 | R08 |
| | avail./NA. | PUR12 | 0.2 / — | 31 / — | Dec 2012 | EG16 |
| | NA./avail. | SVD15 | — / 50 | — / 5 | Apr–Jun 2015 | F19 |
| | NA./avail. | CVAO | — / 30 | — / 7 | Sep, Oct 2017 | T21a |
| $PG_{sw}$/$PG_{aer}$ | avail./avail. | CVAO | 1 / 30 | 16 / 5 | Sep, Oct 2017 | T21b |
| | avail./NA. | AS08 | 0.5 / — | 6 / — | Apr, Jun, Aug 2008 | F11 |





## 4.2 Aircraft observations of organic aerosols

To provide an overview of the model's capability to represent PMOA in remote oceanic regions where ground-based measurements are unfeasible, we compare the simulated PMOA concentrations with aircraft observations over the Arctic, Pacific,
Atlantic, and Southern Oceans. Mass concentrations of organic aerosols (OA) from the Atmospheric Tomography (ATom, https://espoarchive.nasa.gov/archive/browse/atom, last access: 1 August 2024) campaigns of the US National Aeronautics and Space Administration (NASA) are used for the comparison to the aerosol model results. Information regarding the different instruments on board the aircraft, measuring aerosol quantities, can be found at the official website. The aircraft flew mostly over open ocean areas and within the Arctic region. The data is available at a temporal resolution of approximately fifteen
minutes.

Some studies analyzing ATom data indicate that PMOA could significantly contribute in remote oceanic regions to the OA (Pai et al., 2020). Based on Pai et al. (2020), we selected, for the model comparison, the near surface levels from ATom data (heights under 1 km). In an attempt to exclude anthropogenic organic aerosol sources that mostly influence the Northern Hemisphere, we imposed a threshold to OA values and excluded those over $0.2\ \mu g\,m^{-3}$ (measured at standard temperature and pressure con-
ditions; 273 K, 1 atm) as in Pai et al. (2020) the regime's classification. The measured mass concentration of organic aerosols in the submicron aerosol size range in remote regions present the lowest concentration (mostly lower than $0.1\ \mu g\,m^{-3}$), indicating a predominance of local natural marine sources (not shown). Some data inland where primary marine aerosols have no impact were excluded by applying the aforementioned conditions, reducing the dataset to the open ocean or the near coastal regions. Figure 3(a) shows the data grouped by months after applying the aforementioned mentioned conditions. The majority of the
data was measured in October 2017, followed by February of the same year and May 2018 (with a number of observations of 64, 40 and 30). The rest of the cases contain less than 12 observations. Since some of the flight trajectories overlap, the Fig. 3(a) does not visually represent the actual number of samples for some cases. Additionally, to capture regional patterns and to differentiate purely pristine regions from anthropogenic or more dusty polluted areas, the model evaluation was performed for the regions in Figure 3(b). For the evaluation of model results, the hybrid model vertical levels were transformed into pressure
levels and linearly interpolated to the flight horizontal coordinates and altitude where the aerosols were sampled. Since the model temporal resolution is 12 h, we spatially interpolated the flight points, which laid within the 12 h range of the aerosol model. This means that all flight coordinates between 00:00 (12:00) UTC and 12:00 (00:00 of next day) UTC of a certain day are interpolated to the model output corresponding to the same day at 12:00 (00:00 of next day). Then, we derived daily averages of observations and model values and calculated the correlation and normalized model bias for the whole dataset.
Additionally, model results were converted to standard conditions of temperature and pressure to meet the conditions at which the ATom aircraft samples were measured.





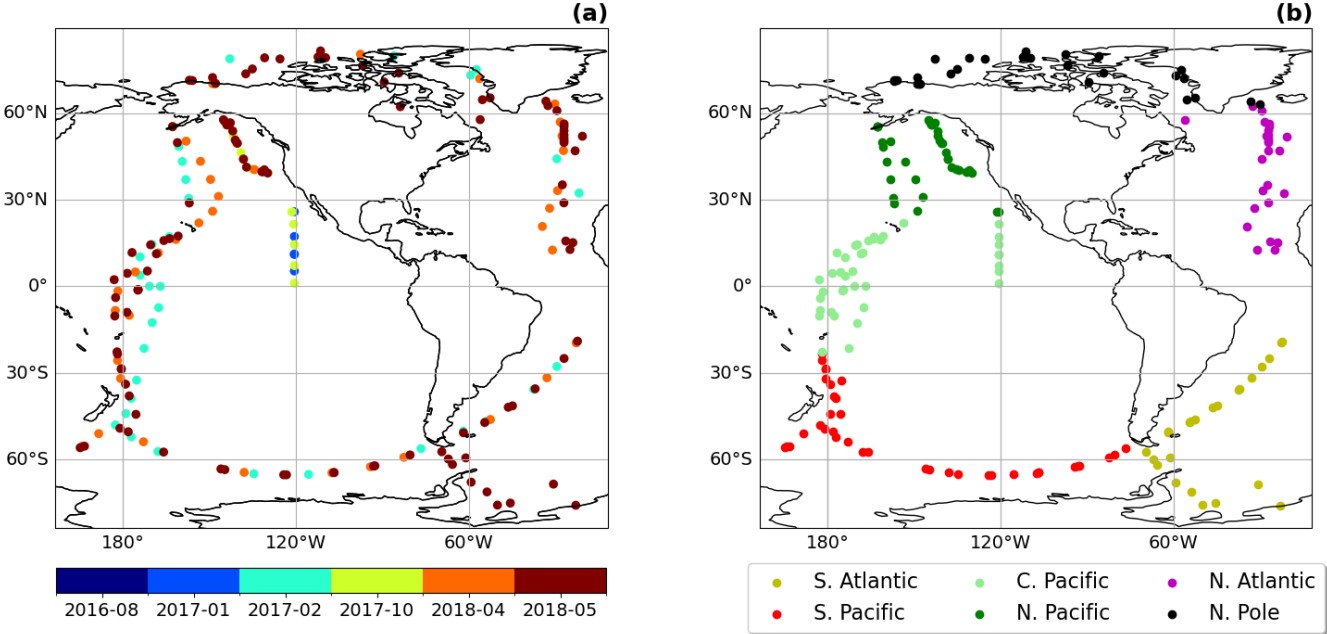

**Figure 3.** (a) Airborne organic aerosol particles grouped by days for diameters smaller than 1 $\mu$m, altitudes below 1 km and concentrations smaller than 0.2 $\mu$g s m$^{-3}$. (b) Colour-coded regions selected for the model evaluation.

## 5 Results and Discussion

### 5.1 Geographical distribution of modelled biomolecules

Based on the FESOM2.1-REcoM3 model data and the calculations in Section 2.3, the three marine key biomolecules at the sea surface have an average spatial distribution as shown in Figure 4(a–c). In addition, Fig. 4(d–f) presents the aerosol organic mass fraction (OMF) calculated with the OCEANFILMS scheme as considered in this study, in an offline mode with the simulated ocean surface concentration as input data.

### 5.1.1 Sea surface concentration of biomolecules

The global distribution of marine biomolecules exhibits distinct patterns for the semi-labile groups PCHO$_{sw}$ and DCAA$_{sw}$, in contrast to the labile PL$_{sw}$ group, due to their resistance to rapid microbial utilization. PCHO$_{sw}$, as the main extracellular product of phytoplankton, has a maximum concentration of up to 8.4 mmol C m$^{-3}$. DCAA$_{sw}$, followed by PL$_{sw}$, have values as high as 2.5 and 1.28 mmol C m$^{-3}$, respectively (see Fig. 4(a, b)).

PCHO$_{sw}$ and DCAA$_{sw}$ show persistently high concentrations over tropical waters (Fig. 4(a, b)). This is linked to the strong
stratification that prevents deep vertical mixing and remineralization. In addition, we also associate these patterns to the carbon-overflow hypothesis (Engel et al., 2004, 2020), in which the carbon exudation increases under nutrient-limiting conditions. In





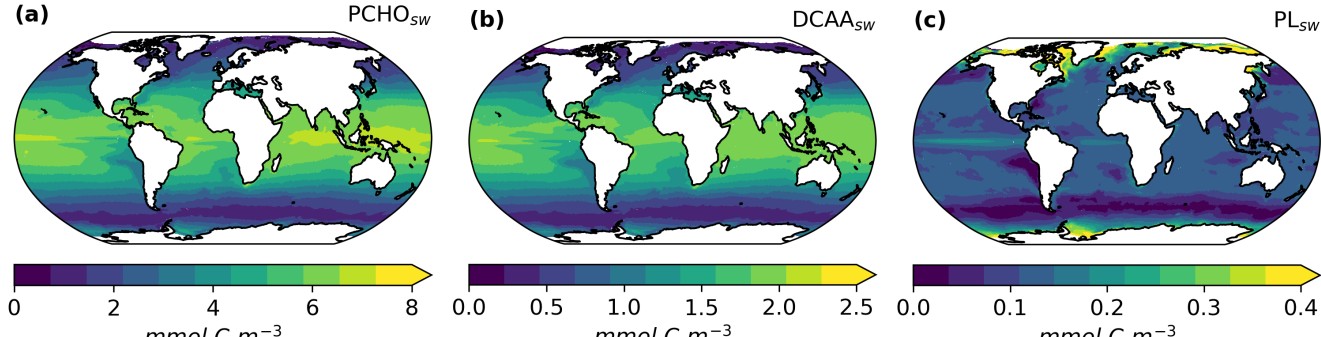

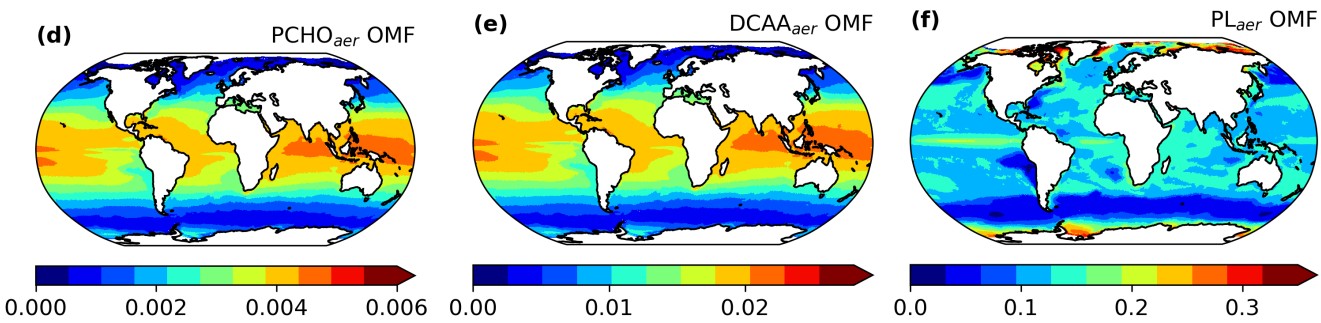

**Figure 4.** Maps of global averaged (a–c) ocean carbon concentration of $PCHO_{sw}$, $DCAA_{sw}$, and $PL_{sw}$ and (d–f) offline computed OMF of $PCHO_{aer}$, $DCAA_{aer}$, and $PL_{aer}$ as a multiannual mean for the period 1990-2019 for sea ice free conditions (SIC < 10 %).

FESOM2.1-REcoM3, nitrogen is indeed the most limiting factor of small phytoplankton in the vast areas of tropical and sub-tropical waters (Schourup-Kristensen et al., 2014; Gürses et al., 2023). For polar regions, on the other hand, the model results have the lowest values during polar night. In the bloom period during northern hemispheric spring, the $PCHO_{sw}$ and $DCAA_{sw}$

concentrations in the Arctic rise to values slightly higher than 8 and 2.5 mmol C m$^{-3}$, respectively (see Fig. C1(d–e). For the Southern Ocean, however, the maximum tends to be lower, with 5.3 mmol C m$^{-3}$ for $PCHO_{sw}$ and 1.6 mmol C m$^{-3}$ for $DCAA_{sw}$ (see Fig. C1(a, b). The distinction can be attributed, among other factors, to the presence of significant river mouths in the Arctic, which serve as a significant source of nutrients for the polar waters that are not present in the Southern Ocean.

The greatest contributions of $PL_{sw}$ (Fig. 4(c)) are found in the equatorial upwelling region and polar waters (see Fig. C1).

Lower values predominate in the subtropical gyres where carbon exudation is solely dominated by the small phytoplankton group. In contrast to the other biomolecules, the values are lowest in the tropics, while higher contributions are found in the Arctic and Antarctic waters during the bloom period (Fig. C1(c, j)). Lastly, the low quantities simulated in the subtropical Pacific off the west coast of South America are common for all biomolecules. For this area, the dissolved inorganic nitrogen





in the FESOM2.1-REcoM3 model has been excessively high compared to observations (Gürses et al., 2023). Additionally, the

high nitrogen concentration also observed in the Southern Ocean could explain the low phytoplankton carbon exudation in this region. For this case, the intracellular nitrogen quota of small phytoplankton is high. As a consequence, the limiting function in Eq. 5 downregulates the carbon excretion. Therefore, the modelled carbon phytoplankton exudation is minimal given the elevated availability of nitrogen in these areas.

In addition to the spatial patterns, the seasonality of biomolecules in the ocean was also analysed by region (Table 6). All quanti-

ties are highest during hemispheric summer for the poles and subtropics (Fig. C1). For June-July-August, the total concentration of all biomolecules is on average $3.72 \pm 1.74 \, \mathrm{mmol\,C\,m^{-3}}$ and $0.55 \pm 0.23 \, \mathrm{mmol\,C\,m^{-3}}$ for the Arctic and Southern Ocean, respectively. During December-January-February the concentration increases in the Southern region ($4.61 \pm 2.7 \, \mathrm{mmol\,C\,m^{-3}}$) and declines in the Arctic Seas $0.68 \pm 0.29 \, \mathrm{mmol\,C\,m^{-3}}$). Whereas the concentrations of $\mathrm{PCHO}_{sw}$ and $\mathrm{DCAA}_{sw}$ drop by about $80\,\%$ during the polar night, $\mathrm{PL}_{sw}$ falls to almost zero (Table 6). The amplitude between seasons is nearly twice

smaller for the subtropics compared to the poles. Nevertheless, total quantities are larger for subtropical waters with values of $5.48 \pm 1.81 \, \mathrm{mmol\,C\,m^{-3}}$ ($3.75 \pm 2.02 \, \mathrm{mmol\,C\,m^{-3}}$) and $5.07 \pm 1.73 \, \mathrm{mmol\,C\,m^{-3}}$ ($2.9 \pm 1.9 \, \mathrm{mmol\,C\,m^{-3}}$) during summer (winter) for the northern and southern high latitudes, respectively. In contrast, seasonal patterns diminish towards the equator ($7.9 \, \mathrm{mmol\,C\,m^{-3}}$), where they are absent due to the intense solar radiation being a limiting factor rather than nutrient depletion (Fig. C1). To our knowledge, there have been scarcely any studies on the surface concentration of marine organic compounds

relevant to aerosols and none on the modelling of the carbon groups presented here. Two examples that closely resemble this work are the studies conducted by Ogunro et al. (2015) and Burrows et al. (2014). Among other groups and bio-indicators of ocean marine biological activity, Ogunro et al. (2015) represented the abundance of polysaccharide-, protein- and lipid-like mixtures. Burrows et al. (2014) based their OCEANFILMS calculations on the macromolecule quantities, similarly derived from the same biogeochemistry model as in Ogunro et al. (2015). These two research studies assumed that the primary source

of total DOC was cell lysis, and based on this assumption, they calculated the concentration of the macromolecular groups at the sea surface. Their computed quantities encompass a broader group than those examined in our study. Polysaccharides were assumed to be equal to the semi-labile dissolved organic carbon pool from a biogeochemical model (Ogunro et al., 2015; Burrows et al., 2014), which is equivalent to the $\mathrm{DOC}_{phy\_ex}$ from FESOM2.1-REcoM3 in our study. With this assumption, other potential sources of polysaccharides are included, leading to higher values than in this work (up to 10-fold) and slightly

different geographical distribution. Nonetheless, similar to our results, the abundance of proteins and polysaccharides is more pronounced in less productive waters compared to high productivity regions (Burrows et al., 2014). As in our case, Burrows et al. (2014) assumed that the protein-like group is composed of a fraction of polysaccharides. Consequently, proteins have the same global distribution and seasonal characteristics as polysaccharides. Finally, the lipid-like mixture in Ogunro et al. (2015) also represents a larger group. In addition, the calculations to estimate this macromolecule include the zooplankton levels and

the rate of phytoplankton disruption by zooplankton grazing. As a consequence, higher values for the concentration at sea surface are found for this group (about 5-fold) compared to $\mathrm{PL}_{sw}$ in the present study. Nonetheless, the ocean distribution of $\mathrm{PL}_{sw}$ agrees reasonably well with that presented by Burrows et al. (2014). Regardless of the considerations assumed to compute the biomolecules in the ocean, our approach adequately depicts the abundance of the biomolecules in the ocean that are relevant





to the aerosols. The seasonal patterns modelled here also agree to those in Burrows et al. (2014) and polysaccharide group is
most frequently represented, followed by amino acids and lipids.

**Table 6.** Carbon concentration of marine biomolecules for December-January-February (DJF) and June-July-August (JJA) as a multiannual monthly mean for the period 1990-2019 averaged over the polar regions (Arctic Ocean $60°N - 90°N$, Southern Ocean $63°S - 90°S$), Northern and Southern Subtropics ($23°N - 60°N$ and $23°S - 60°S$) and Equator ($23°N - 23°S$). Values are in $\mathrm{mmol\,C\,m^{-3}}$. In parentheses, the multi-seasonal and regional standard deviation.

| Regions | $PCHO_{sw}$ | | | $DCAA_{sw}$ | | | $PL_{sw}$ | | |
|---|---|---|---|---|---|---|---|---|---|
| | JJA | | DJF | JJA | | DJF | JJA | | DJF |
| Arctic Ocean | 2.43 (1.20) | > | 0.52 (0.21) | 0.73 (0.36) | > | 0.16 (0.07) | 0.57 (0.36) | > | $2.4 \times 10^{-6}$ |
| Southern Ocean | 0.41 (0.19) | < | 3.31 (2.0) | 0.13 (0.06) | < | 0.99 (0.6) | $1.5 \times 10^{-7}$ | < | 0.31 (0.16) |
| N. Subtropics | 4.09 (1.41) | > | 2.85 (1.53) | 1.22 (0.42) | > | 0.85 (0.46) | 0.16 (0.13) | > | 0.05 (0.05) |
| S. Subtropics | 2.21 (1.42) | < | 3.81 (1.31) | 0.66 (0.43) | < | 1.14 (0.4) | 0.03 (0.04) | < | 0.13 (0.07) |
| Equator | 5.97 (0.71) | $\approx$ | 6.0 (0.63) | 1.78 (0.21) | $\approx$ | 1.79 (0.19) | 0.11 (0.03) | = | 0.11 (0.03) |

## 5.2 Aerosol organic mass fraction

The organic mass fraction in aerosols depends on the distribution of marine biomolecule concentration in the ocean (Fig. 4(d–f)). In contrast to the abundance of organic groups in seawater, the OMF of polar lipids is significantly higher than that of the
other groups during the hemispheric summer (see Fig. C2. Contributions can be as high as 0.44. In contrast, for $PCHO_{aer}$ and $DCAA_{aer}$, OMF values are low, reaching a maximum of 0.004 and 0.02, respectively (Fig. 4(d–f)). The disproportional enrichment observed in the aerosol phase is explained by the aforementioned characteristics of the surface affinity of the main biomolecule groups in seawater (Sect. 2.1). Lipids are highly active surfactants whose surface affinity favours their transfer to the aerosol phase. Consequently, the OMF of $PL_{aer}$ is two to three orders of magnitude greater than that of $PCHO_{aer}$ and
$DCAA_{aer}$ (Fig. 4(d, e)) and these high values persist globally.

Among the biomolecules, $PL_{aer}$ OMF is the group showing the most pronounced seasonal patterns with values generally decreasing towards the Equator during the hemispheric summer (Table 7 and Fig. C2(c, f, j, m)). $PCHO_{aer}$ and $DCAA_{aer}$ OMF values, on the other hand, remain uniform across seasons for subtropical and equatorial areas (see Fig. C2). For the polar regions, the abundance of all organic groups in aerosols has a clear seasonality, with strong changes for $PL_{aer}$ group
(see Fig. C2 (d–j). These seasonal characteristics are caused by an increase in marine primary production as light limitation decreases at the end of the winter. With melting sea ice, light is available in ice free areas or passes through the thin ice triggering the photosynthesis of phytoplankton. Once nutrients present in seawater are consumed and the polar night sets in, the biological productivity and atmospheric contribution of marine organics are significantly diminished, especially for $PL_{aer}$. As expected, the responses of the various groups mirror those in Burrows et al. (2014). Nevertheless, based on the fundamental





**Table 7.** OMF of biomolecule groups in aerosols for the same regions and seasons in Table 6.

| Regions | $PCHO_{sw}$ | | | $DCAA_{sw}$ | | | $PL_{sw}$ | | |
|---|---|---|---|---|---|---|---|---|---|
| | JJA | | DJF | JJA | | DJF | JJA | | DJF |
| Arctic Ocean | 0.001 (0.001) | > | 0.0004 (0.0002) | 0.006 (0.002) | > | 0.002 (0.001) | 0.316 (0.090) | > | $3.35 \times 10^{-6}$ |
| Southern Ocean | 0.0003 (0.0001) | < | 0.002 (0.001) | 0.001 (0.001) | < | 0.009 (0.005) | $2.2 \times 10^{-7}$ | < | 0.237 (0.080) |
| N. Subtropics | 0.003 (0.001) | > | 0.002 (0.001) | 0.013 (0.004) | > | 0.010 (0.005) | 0.158 (0.051) | > | 0.059 (0.049) |
| S. Subtropics | 0.002 (0.001) | < | 0.003 (0.001) | 0.008 (0.005) | < | 0.012 (0.004) | 0.039 (0.044) | < | 0.134 (0.047) |
| Equator | 0.004 (0.0004) | = | 0.004 (0.0004) | 0.019 (0.002) | = | 0.019 (0.002) | 0.124 (0.026) | ≈ | 0.126 (0.025) |

distinctions among the organic classes analysed, between the two studies, our OMF values are comparatively smaller yet still comparable to their findings.

## 5.3 Evaluation of modelled biomolecules at the ocean surface

A comparison of simulated biomolecule concentrations with their measured counterparts in seawater is presented in this section. Each group is analysed against its analogous group from observations (see Sect. 4.1), supported by a discussion of the factors associated with model uncertainties. Figure 5 shows the ocean concentration of modelled and observed quantities for the locations in Fig. 2, for which ocean measurements were available (Table 5). Note that modelled $PCHO_{sw}$ and semi-labile $DCAA_{sw}$ represent a fraction of measured $DCCHO_{sw}$ and $DCAA_{sw}$, whereas observed $PG_{sw}$ forms part of the $PL_{sw}$ group.

### 5.3.1 Carbohydrates

In Fig. 5, the observations of $DCCHO_{sw}$ concentration (blue boxes) tend to be lower at the northern and southern stations compared to the subtropics. Median values are at the lowest limit for the West Antarctica Peninsula (WAP) ($0.7 \pm 0.8$ mmol C m$^{-3}$) followed by the North-Atlantic Ocean (NAO) ($1.3 \pm 0.4$ mmol C m$^{-3}$). In contrast, the campaigns conducted in the Peruvian Upwelling Region (PUR12 and PUR17) recorded median concentrations of $4.8 \pm 3.8$ mmol C m$^{-3}$ and $3.9 \pm 1.3$ mmol C m$^{-3}$ in the years 2012 (PUR12) and 2017 (PUR17), respectively. Lastly, values from the Cape Verde Atmospheric Observatory (CVAO) in the subtropical Atlantic are within the range of the other stations ($2.4 \pm 0.4$ mmol C m$^{-3}$) and at the lower end of the PUR12 and PUR17 stations.

The model (coral boxes in Fig. 5) can capture most of the regional variations seen in observations. Quantities tend to be higher for NAO compared to WAP. The lowest $PCHO_{sw}$ concentration occurs at the southern station (median of $1.0 \pm 0.47$ mmol C m$^{-3}$). This may be attributable to the limitations of the important nutrient iron in the Antarctic region (Gürses et al., 2023). Interestingly, the variability for WAP is better represented, although at a higher value than observations. Greater quantities are found at PUR12 and PUR17, where nutrients are transported from the seabed to the surface. Here, the modelled $PCHO_{sw}$ concentration is in good agreement with observed $DCCHO_{sw}$, with a modelled median of $4.1 \pm 0.39$ mmol C m$^{-3}$ and $3.8 \pm 0.16$ mmol C m$^{-3}$ for PUR12 and PUR17, respectively. The lowest normalized mean bias (NMB) is detected for PUR17



(0.09), whereas values are slightly overestimated for WAP (NMB = 0.64) and underestimated for PUR12 (NMB = -0.27). The significant variability observed in PUR12 is not adequately captured by the model. The sampling depth may provide an explanation for this. The quantities would likely be greater for PUR12 if the water samples had been collected at 20-cm depth, in contrast to 1–3 m for PUR17. Maßmig and Engel (2021) found that concentrations of DCCHO and DCAA decrease with depth in this region. Note that FESOM2.1-REcoM3 vertical resolution is coarser and cannot resolve the processes within the SML solely including an average of the upper 5m depth. Additionally, the model data used are a volume-weighted mean over the upper 30 m, and it is unlikely that subsurface water biomolecule abundances are accurately represented. Nonetheless, our results indicate that the modelled surface carbon concentrations of the biomolecules are in reasonably good agreement with the observations.

On the other hand, the modelled concentrations are about four times higher than the measurements for NAO ($5.75 \pm 2.48$ mmol C m$^{-3}$) and 2.5 times higher for CVAO ($6.12 \pm 0.01$ mmol C m$^{-3}$). NMB values for these locations are 2.04 and 1.42 for the northern and tropical sites, respectively. For these sites, the sampling size is relatively small (n = 3) and the observations may not fully represent DCCHO$_{sw}$ in the region. Nevertheless, the overestimation by the model could be explained by the carbon-overflow hypothesis (Engel et al., 2004, 2020), which states that carbon exudation is increased under nitrogen-limiting conditions. More-over, bacteria are abundant in oligotrophic provinces like at CVAO and likely consume DOC, a process that is not explicitly represented in FESOM2.1-REcoM3 (Gürses et al., 2023).

In addition to the factors mentioned, we also link the overestimation in polar regions to the fixed fraction of exuded DOC that is generalized to all phytoplankton growth phases. Phytoplankton acidic polysaccharides excretion tend to be stronger during the post-bloom period compared to the phytoplankton growth phase. The amount of PCHO$_{sw}$ exuded could be half the of that currently used in the model representative of the post-bloom phase (Schartau et al., 2007).

### 5.3.2 Amino acids

The highest concentration of sampled DCAA$_{sw}$ was found for CVAO with a median of 4.1 mmol C m$^{-3}$. On the other hand, the levels for PUR12, Stony Brook (SB) and the Northwest Atlantic Ocean site (NWAO) are closely aligned with median values of $1.99 \pm 0.83$, $2.2 \pm 1.48$ and $3.04 \pm 2.21$ mmol C m$^{-3}$, respectively. Among them, Stony Brook has the widest range in the concentration. The higher variability is attributed to a substantial number of year-round measurements conducted at this location. Interestingly, the Peruvian Upwelling Region and the northwest Atlantic Ocean show a similar variability, although they were measured in different individual months (December 2012 and June 2001, respectively). Conversely, the subtropical stations, the oligotrophic site of SATL and the Mediterranean station show significantly less variability during the sampling period September to October, but also have fewer data samples compared to the other stations. Concentrations are relatively lower for WMED ($1.62 \pm 0.27$ mmol C m$^{-3}$) than for SATL ($1.9 \pm 0.29$ mmol C m$^{-3}$).

The computed quantities are within the same range for all stations. WMED and SATL are properly represented with median values close to observations ($1.66 \pm 0.01$ and $1.78 \pm 0.05$ mmol C m$^{-3}$, respectively) and a low model bias (NMB = 0.10 for WMED and -0.07 for SATL). For the other locations, the estimated concentrations are confined to the lower quartile of the observations. The variability for subtropical sites is also apparent in the modelled DCAA$_{sw}$; however, it is not properly cap-





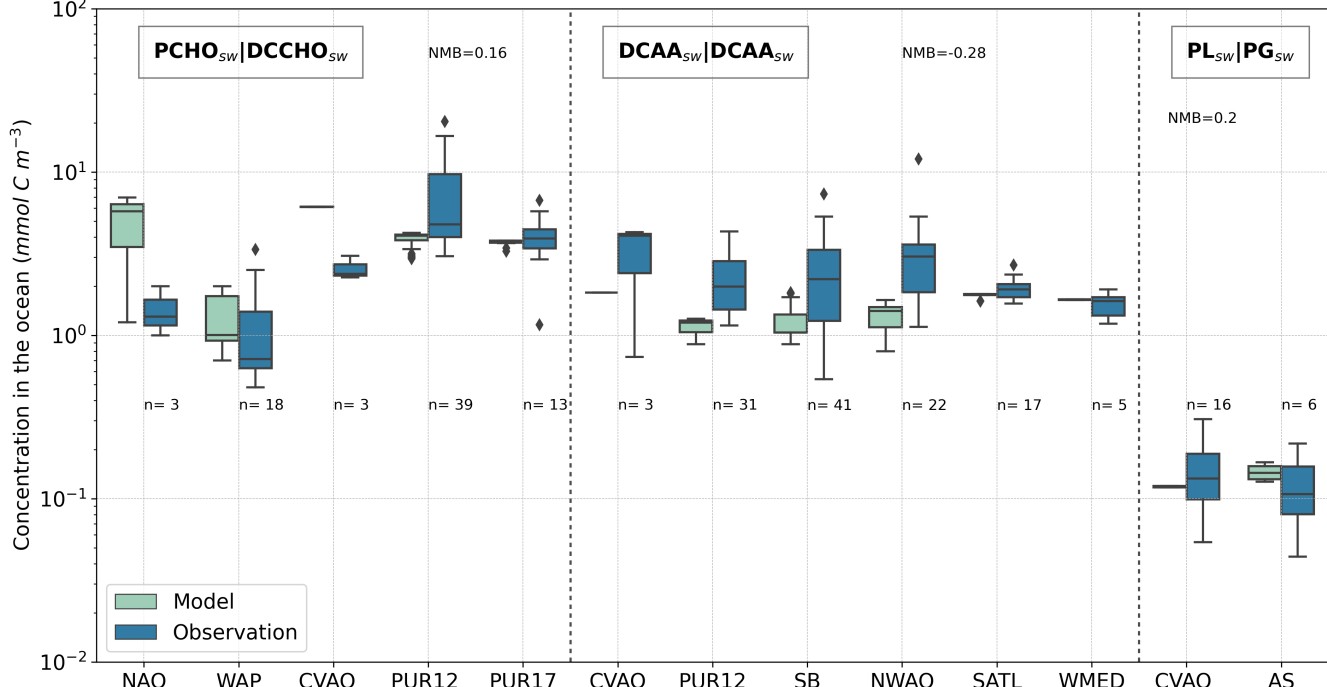

**Figure 5.** Box plot of carbon concentration in seawater of modelled $PCHO_{sw}$, $DCAA_{sw}$ and $PL_{sw}$, and measured $DCCHO_{sw}$, $DCAA_{sw}$ and $PG_{sw}$ for the locations in Fig. 2 (see also Table 2 and Table 4 for the compounds and location abbreviations, respectively). Blue boxes represent the bulk water samples (Table 5) and coral boxes are the modelled biomolecule concentration interpolated to the coordinates where the water samples were collected. Normalized mean bias (NMB) is included for each group; number of observations (n) is included for all sites. The formula to calculate NMB may be found in Table A1.

tured. The normalized model bias ranges between -0.42 and -0.29 for these sites, indicating an underestimation of the observed values. Apparently, regional patterns are also found in the simulated biomolecules. For example, as for the observations, CVAO has the largest values and median concentrations of modelled $DCAA_{sw}$ for NWAO ($1.4 \pm 0.28$ mmol C m$^{-3}$) tend to be higher

535    than for PUR12 ($1.2 \pm 0.12$ mmol C m$^{-3}$) and SB ($1.04 \pm 0.29$ mmol C m$^{-3}$).

The differences between the modelled and measured values are determined primarily by the approach used to calculate $DCAA_{sw}$. This biomolecule group was derived from $PCHO_{sw}$ concentration. Hence, detailed processes relevant to the production of amino acids in seawater are not represented here. Furthermore, as for carbohydrates, the concentrations of $DCAA_{sw}$ tend to be greater near the surface (Maßmig and Engel, 2021). Therefore, the underestimations found for most cases could also

540    be determined by the subsurface sampling depth, which was a maximum of 30 cm. Despite this, the modelled quantities agree reasonably well with the observed $DCAA_{sw}$.



### 5.3.3 Polar lipids

The observed $PG_{sw}$ concentrations are on average $0.13 \pm 0.08$ mmol C m$^{-3}$ for CVAO and slightly lower for the Adriatic Sea (AS) ($0.11 \pm 0.06$ mmol C m$^{-3}$). Our modelled results show higher concentrations for AS ($0.14 \pm 0.02$ mmol C m$^{-3}$) than for CVAO ($0.12$ mmol C m$^{-3}$, with a larger NMB for AS ($0.66$). Note that $PG_{sw}$ may not fully represent all polar lipids in seawater. Nevertheless, the model values are in good agreement with the observed $PG_{sw}$, with the lowest model biases (NMB = 0.2) compared to the other groups.

The observations indicate that the presence of the $PG_{sw}$ group exhibits a strong correlation with the presence of phytoplankton (Triesch et al., 2021b). Therefore, the model estimates are closely aligned with the observed fraction of lipids produced by phytoplankton excretion. Nevertheless, the model cannot reproduce the variability among the tropical stations. The monthly median values for this case remain within the same range ($0.11 \pm 0.03$ mmol C m$^{-3}$ in September and $0.15 \pm 0.1$ mmol C m$^{-3}$ in October); however, there are notable inter-month changes that cannot be captured by the monthly means of the model.

Despite the fact that each biomolecule is analyzed against broader measured groups in the ocean, this comparison serves as an indication of how effectively the modelled biomolecules are represented in terms of magnitude and geographic distribution.

In summary, model uncertainties depend on the considerations used to compute biomolecules, in which we neglected processes involved in their production or consumption. Additionally, the temporal and spatial resolution of the model is a source of uncertainty. Firstly, the monthly model values cannot represent changes within the same month (e.g., PUR12 and CVAO). Therefore, in certain instances, minor variations within a month may not be discernible, resulting in relatively homogeneous values and minimal standard deviations for the modelled quantities, such as CVAO, SATL, and WMED. Secondly, a common feature for tropics and subtropics is the coarser resolution of the non-uniform FESOM2.1-REcoM3 model mesh (Schourup-Kristensen et al., 2018) which could, at some extent, decrease the model accuracy for those regions. Lastly, the averaged values over the first 30 m of the model output agree better with observed biomolecule groups when the sampling depth was 1 m or deeper. Furthermore, for some stations (e.g., CVAO) the number of water samples was small, often within the same month, may not be statistically representative of the existent variable conditions of the biomolecules for the region.

Regardless of the model biases, the calculated quantities lay within the same order of magnitude compared to observations for all cases. The different abundances of biomolecule groups in the ocean are well captured. The lipid group is the least abundant, with a concentration at least one order of magnitude lower than that of carbohydrates and amino acids.

## 6 Global atmospheric simulations of PMOA

We have already discussed in detail the geographical distribution and seasonality of marine biomolecules. Their concentrations at the sea surface serve as boundary conditions for the aerosol model. The results of the global aerosol simulations are presented in the following sections. We included an analysis of global mean burden and emission mass flux in the context of previous PMOA modelling studies. Furthermore, a comprehensive model evaluation against aerosol measurements is presented.



## 6.1 Emission and transport characteristics

The global mean emission and burden values of marine aerosol are summarized in Table 8 for all organic species and sea salt. In addition, PMOA quantities are given as totals of the marine organic aerosol groups. Global emissions and burden are mainly governed by $PL_{aer}$, representing about 1 % of the sea salt emission by mass. This group accounts for 87.2 % of PMOA, whereas $PCHO_{aer}$ and $DCAA_{aer}$ make up 2.3 % and 10.5 %, respectively. Since hygroscopicity parameters are assumed to be identical and all groups are emitted into the same aerosol mode of the HAM model, their contribution to the total burden

remains relatively unchanged compared to the emissions.

The global emission values modelled in this study total 13.6 Tg yr$^{-1}$, which is within the range of previous studies that vary between 9 and 27 Tg yr$^{-1}$ (Meskhidze et al., 2011; Huang et al., 2018; Zhao et al., 2021). Similarly, a total mean burden of 0.068 Tg agrees with other studies, ranging from 0.048 to 0.097 Tg (Huang et al., 2018; Zhao et al., 2021; Burrows et al., 2022). Lastly, the ratio of PMOA to SS, shows clearly the dependency on the sea salt emission source function used. This

percentage tends to be slightly larger in our case (1.24 %) compared to Zhao et al. (2021)(0.67 %) and Meskhidze et al. (2011) (0.7 %). We believe this is a consequence of the considerably larger emission fluxes and burden of SS in their results compared to our work.

The difference with other studies may be caused by the parameterization used to calculate PMOA. While we apply a physically based scheme to capture the bubble bursting process and the transfer of marine organics to the aerosols, Meskhidze et al.

(2011) employed only a chl-$a$ based approximation. Nevertheless, despite the different approaches to computing PMOA, our values are closer to the study by Huang et al. (2018), given the similarities in terms of model configuration. This indicates that discrepancies with other modelling studies are largely attributable to the sea salt emission scheme and the dynamics, transport, and deposition in the respective atmospheric aerosol transport models, rather than the PMOA emission scheme.

Surface wind is probably the most important factor controlling PMOA emissions. The PMOA representation depends directly

on wind speed (Gantt et al., 2011) or on a sea spray emission source function (Meskhidze et al., 2011; Zhao et al., 2021). Sea salt emissions are typically parameterized in relation to the 10 m wind speed and/or as a function of the sea surface temperature (Gong, 2003; Mårtensson et al., 2003; Long et al., 2011).

For a more in-depth understanding of the driving processes that control the emission, dispersion, and removal of marine organic aerosol, we examine the connection among multiple model variables. In our study, the PMOA surface emission fluxes

(Fig. 6(a)) are mainly driven by surface wind. Their geographic distribution is strongly linked to that of sea salt. Nonetheless, in regions where sea salt emissions are relatively low (e.g., high latitudes over 60 degrees of each hemisphere during summer), the distribution of marine organic aerosol is primarily dominated by the elevated biological activity during the bloom period. The strongest emission fluxes occur in the North Pacific, North Atlantic, and Indian Oceans with maximum values of 8.20, 8.77 and 8.69 ng m$^{-2}$ s$^{-1}$, respectively. The total mean emission flux for North Pacific waters tends to be 35 % larger than that

for the North Atlantic. For the Southern Ocean, the total mean emission flux is low (0.35 Tg yr$^{-1}$) and values in neighbouring waters (South Atlantic, Indian Ocean, and Pacific oceans) are 5 to 10 times higher.

The burden exhibits a similar behaviour to the emission flux. Quantities for the Pacific Ocean (0.016 to 0.02 Tg) remain larger



**Table 8.** Global mean emission flux and burden of marine species and sea salt.

| Species | Emission ( Tg yr$^{-1}$) | Burden (Tg) | PMOA / SS emission (%) |
|---|---|---|---|
| PCHO$_{aer}$ | 0.31 | 0.002 | 0.03 |
| DCAA$_{aer}$ | 1.43 | 0.007 | 0.13 |
| PL$_{aer}$ | 11.90 | 0.060 | 1.08 |
| Total PMOA | 13.6 | 0.068 | 1.24 |
| SS | 1100.00 | 3.73 | — |

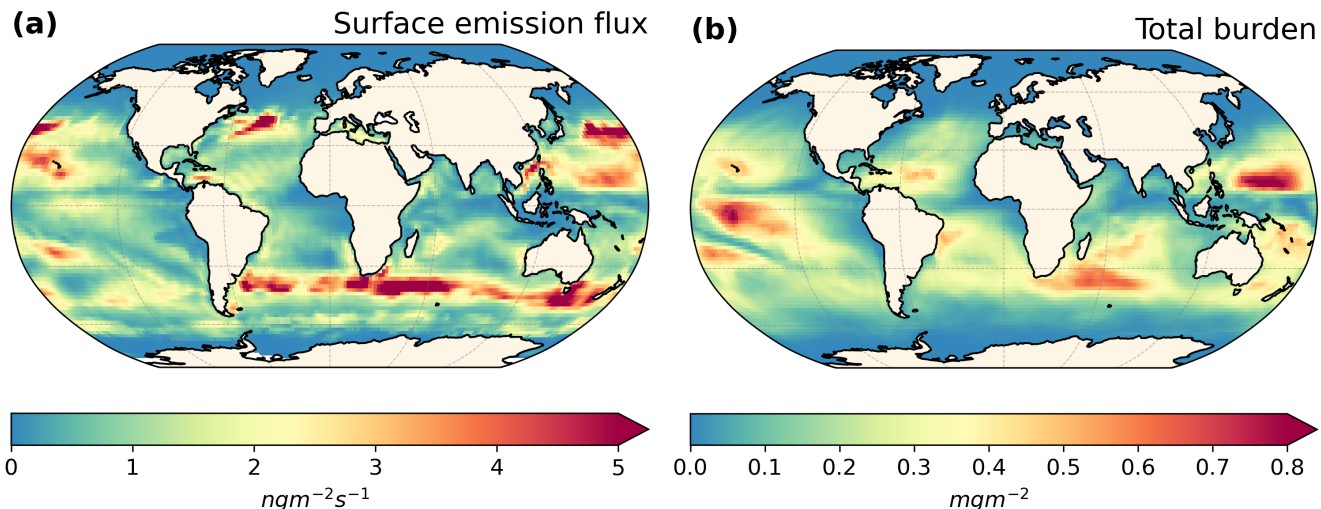

**Figure 6.** Maps of PMOA means of (a) global surface emission mass flux and (b) total burden for the simulated ECHAM6.3–HAM2.3 period 2009–2019.

than for the Atlantic Ocean (0.006 to 0.009 Tg). Unlike the Atlantic, the difference in production increases between North and South Pacific and is reversed for the burden when compared to the emissions. Such differences, in which the highest emissions

do not coincide with greater burdens, are caused by the transformation and transport processes that the aerosols undergo in the model. Once emitted, aerosols are advected and driven by the general atmospheric circulation. If they are not scavenged by wet deposition mechanisms, they will remain longer in the atmosphere. These processes regulate the total aerosol burden, which peaks in regions with lower surface emissions (e.g., Equatorial Pacific). In regions with high biological productivity, such as the North Atlantic and North Pacific oceans, the main removal mechanism of PMOA is wet deposition, likely due to large-scale

precipitation. In both hemispheres, the wet removal is very efficient. Especially in the Southern Ocean, for instance, the mean total burden is with 0.0005 Tg nearly 30 times lower than in the Indian Ocean. Lastly, the emissions and burden in the Arctic are among the lowest by approximately two and three orders of magnitude in comparison to the North Atlantic waters.





## 6.2 Species-wise evaluation

In this section, we present the results of a comprehensive analysis of species-resolved offline computed OMF and ECHAM6.3–HAM2.3
aerosol concentration against observations. The modelled aerosol organic mass fraction is the result of applying OCEAN-
FILMS with the ocean biomolecule quantities as input data. As a reminder, the OMF depicts the transition from marine organic
material to the aerosol phase for each group, and it is entirely free of meteorological influences. In contrast, aerosol concen-
tration is strongly affected by wind stress, which triggers sea salt emission in the aerosol model. Hence, we have included the
model evaluation against sea salt aerosol concentration for the stations where observations of marine organics are available.
Sodium amounts from observations were used to calculate sea salt concentrations, as explained in Sect. 4.1.

For comparison with measurements of marine organic aerosols, we account on the submicron aerosol concentration and
estimated OMF from observations. For evaluating the model aerosol quantities $PCHO_{aer}$, $DCAA_{aer}$ and $PL_{aer}$, measured
$CCHO_{aer}$, $CAA_{aer}$ and $PG_{aer}$ were selected accordingly. In addition, measured organic mass (OM) concentrations are com-
pared with the modelled total PMOA. Simulated and observed quantities for various stations were compiled into a multi-panel
box plot in Fig. 7. A detailed description of the observational data and the discussion of our model results for each group is
provided below.

Note that the model evaluation is challenging, since the coarse horizontal and vertical spatial resolution of ECHAM6.3–HAM2.3
is a major limitation when we compare our results to observational data. Uncertainties in model results arise from comparing
interpolated grid-averaged data to point measurements, which can lead to significant spatial sampling errors, especially at
coarser resolutions. These errors may decrease with higher grid resolutions but are still influenced by the assumption of aerosol
field homogeneity within the grid cell. This assumption is problematic in areas with inhomogeneous surfaces, such as coast-
lines, where air masses change from ocean to land (Vignati et al., 2001). Moreover, turbulent transport and boundary layer
height are parameterized as in other global aerosol-climate models. Therefore, variable wind conditions over height and time
that influence the aerosol detection (measurement heights of less than 60 m) cannot be explicitly resolved by our model.


### 6.2.1 Carbohydrates

The measured $CCHO_{aer}$ mass fractions in aerosols are found to be within the same range for all stations (blue boxes in Fig. 7
first panel). The OMF measurements range between $3.4 \times 10^{-3} \pm 3.7 \times 10^{-3}$ for CVAO and $3.9 \times 10^{-3} \pm 3.3 \times 10^{-3}$
$(2.4 \times 10^{-3} \pm 1.4 \times 10^{-3})$ for NAO (WAP). Similarly, measured carbohydrate concentrations within submicron aerosol par-
ticles (dark pink boxes in Fig. 7) are also slightly lower for CVAO $(1.1 \times 10^{-3} \pm 1.1 \times 10^{-3} \mu g\,m^{-3})$ compared to the
northern and southern sites $(1.7 \times 10^{-3} \pm 1.1 \times 10^{-3} \mu g\,m^{-3}$ and $2.1 \times 10^{-3} \pm 2.4 \times 10^{-3} \mu g\,m^{-3}$, respectively).

In contrast, $PCHO_{aer}$ OMF obtained from the offline calculation (coral boxes in Fig. 7) is higher for the tropical station
$(4.1 \times 10^{-3})$ than for NAO $(1.6 \times 10^{-3})$ and WAP $(7 \times 10^{-4})$ with respect to the observations. Nonetheless, NAO and
CVAO OMF values are within the same order of magnitude of observations. The discrepancies are significant for WAP fol-
lowed by NAO, with a negative mean model bias of -0.002. The simulated concentrations (light pink in Fig. 7) are also



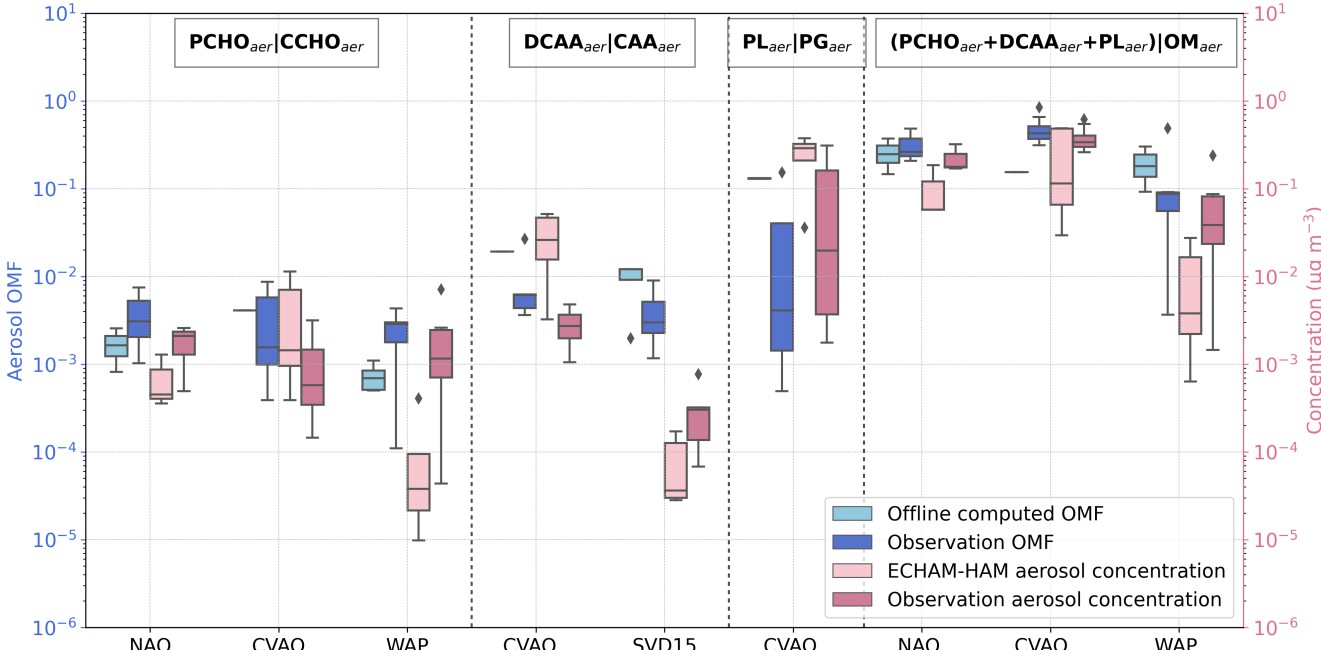

**Figure 7.** Box plot of the species-resolved, offline computed OMF and ECHAM6.3–HAM2.3 simulated concentrations in contrast to the measured values. Submicron aerosols $PCHO_{aer}$, $DCAA_{aer}$ and $PL_{aer}$ are compared to $CCHO_{aer}$, $CAA_{aer}$ and $PG_{aer}$ respectively for the stations in Fig. 2 (see also Table 2 and Table 4 for the compounds and location abbreviations, and Table 5 for detailed information regarding the measurement data). Total modelled marine aerosol ($PCHO_{aer}+DCAA_{aer}+PL_{aer}$) is compared to observations of aerosol organic mass (OM) concentration. Dashed lines separate each group and the names tags indicate the aerosol precursor in the ocean separated by a straight bar from the measured aerosol components selected for the evaluation. Coral and light pink colours mark modelled values, whereas blue and dark pink indicate observations of OMF and aerosol concentration in $\mu g\,m^{-3}$ respectively.

lower than the observations for high latitude sites. As a result, the aerosol concentrations are also underestimated for NAO ($4.5 \times 10^{-4} \pm 5 \times 10^{-4}\ \mu g\,m^{-3}$) and WAP ($3.8 \times 10^{-5} \pm 1.4 \times 10^{-4}\ \mu g\,m^{-3}$), while higher values than the observations were modelled for CVAO ($1.4 \times 10^{-3} \pm 4.3 \times 10^{-3}\ \mu g\,m^{-3}$). It is likely that the slight overestimation for CVAO is due to the relatively higher concentration of this biomolecule in seawater. Interestingly, there is no similar response for NAO or WAP,

for which the ocean biomolecules were over-represented by the model or in good agreement with the water samples. Possible causes for this contradicting pattern can be explained as follows.

Firstly, NAO and WAP, aerosols were sampled while the ship was in motion. Therefore, aerosol samples were influenced by a much greater number of ocean areas than could be covered by surface seawater sampling. Secondly, relevant processes that favour the transfer of carbohydrates to the air are not included in the OCEANFILMS version used in this study. The

under-representation of the organic fraction and concentration of carbohydrates in marine aerosols by the model seems to be a limitation of the monolayer Langmuir model, which neglects molecule interactions (Burrows et al., 2014). In addition to the





aerosol measurements included in the present work, other studies employing different detection techniques have quantified even higher organic enrichment in aerosols attributed to carbohydrates (Russell et al., 2010; Frossard et al., 2014). Co-adsorption mechanisms facilitate the transfer to the aerosol phase of less surface-active compounds, such as polysaccharides or proteins,

which are less enriched at the sea surface microlayer (SML). This occurs when such groups attach to the surfactants already coating the air bubble due to ionic interactions (Burrows et al., 2016; Hasenecz et al., 2019; Link et al., 2019).

Furthermore, given our model assumption, $PCHO_{aer}$ is a fraction of $CCHO_{aer}$. Therefore, modelled dissolved acidic polysaccharides may not represent the total measured combined carbohydrates group but rather a fraction. Ultimately, given that aerosols are not collected under controlled conditions, concentrations may be affected by newly released polysaccharides by

bacteria in the atmosphere (Zeppenfeld et al., 2021, 2023). Hence, $CCHO_{aer}$ concentrations may also originate from secondary sources other than the primary oceanic sea spray emissions.

In addition to the aforementioned factors, the negative biases found in aerosol concentration for WAP (-0.002 $\mu g\,m^{-3}$) and NAO (-0.001 $\mu g\,m^{-3}$) can be associated with an underestimation of the simulated sea salt concentrations (Fig. 8). Their magnitudes are at least an order of magnitude smaller, with a mean bias of -0.61 $\mu g\,m^{-3}$ and -0.2 $\mu g\,m^{-3}$ for WAP and NAO,

respectively. Sensitivity experiments with the ECHAM6.3–HAM2.3 model performed by Huang et al. (2018) demonstrate that when PMOA is emitted together with sea salt as sea spray, PMOA emissions are particularly sensitive to the sea salt source function selected.

### 6.2.2 Amino acids

The organic mass fractions of free combined amino acids ($CAA_{aer}$) from the measurements (Fig. 7 second panel) lay within

the same range as for $CCHO_{aer}$. The median value of OMF obtained from observations is $4 \times 10^{-3} \pm 3 \times 10^{-3}$ for Svalbard station, whereas over twice higher for CVAO ($8.75 \times 10^{-3} \pm 8.8 \times 10^{-3}$). Interestingly, model quantities are about one order of magnitude larger than measurements. For the northern station, the modelled OMF median goes up to, $1.2 \times 10^{-2} \pm 4.4 \times 10^{-3}$ and for CVAO, $0.2 \times 10^{-2} \pm 5.7 \times 10^{-5}$. The simulated OMF values of $DCAA_{aer}$ tend to overestimate observations significantly for the tropical station, with a bias of 0.01 $\mu g\,m^{-3}$. However, for Svalbard, model results are at the upper end of

observation with smaller differences (0.005 $\mu g\,m^{-3}$).

In contrast, the observed concentration of aerosol exhibits distinct patterns. Firstly, CVAO reported quantities of $2.8 \times 10^{-3} \pm 1.4 \times 10^{-3}$ $\mu g\,m^{-3}$, whereas data collected in Svalbard remain smaller with a median of $0.3 \times 10^{-3} \pm 0.3 \times 10^{-3}$ $\mu g\,m^{-3}$. The remarkable differences in OMF are obscured by the higher mean sodium concentration in CVAO, which compensates for the high occurrence of $CAA_{aer}$. The model nicely captures such regional characteristics for these stations. The concentration

simulated at CVAO is $2.6 \times 10^{-2} \pm 1.9 \times 10^{-2}$ $\mu g\,m^{-3}$, which has a positive bias of 0.03 $\mu g\,m^{-3}$ that is amplified as sea salt is overestimated in our model for this site (CVAO_AA in Fig. 8 with a bias of 1.17 $\mu g\,m^{-3}$).

For the measurement campaign on Svalbard in 2015, the model bias is negative ($-3 \times 10^{-4}$ $\mu g\,m^{-3}$) and the median concentration ($3.6 \times 10^{-5} \pm 7 \times 10^{-5}$ $\mu g\,m^{-3}$) is four times smaller than the observations. This is related to a 10-times lower sea salt content (SVD15 in Fig. 8). It appears that, the slightly higher OMF compared to the observations does not compensate for

the underestimation of sea salt (bias=$-0.1$ $\mu g\,m^{-3}$)), which we consider to be the main reason for low $DCAA_{aer}$ aerosol



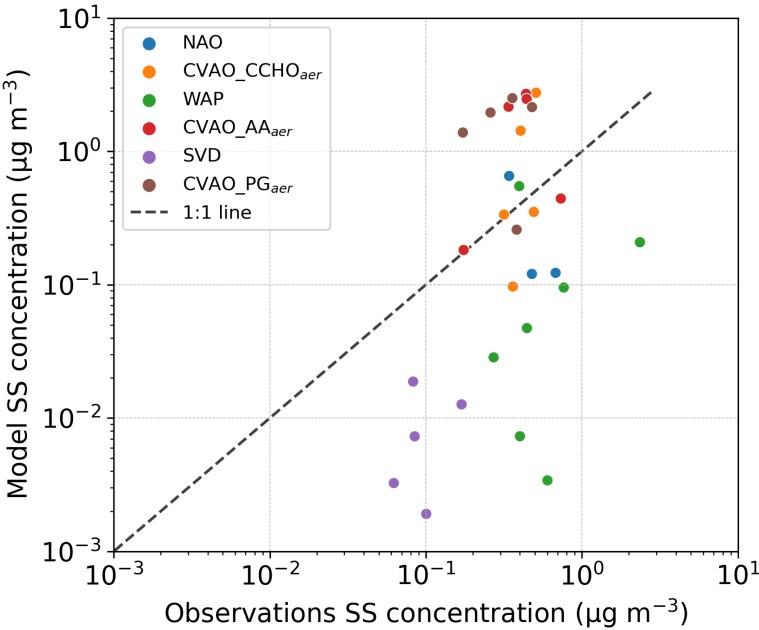

**Figure 8.** Scatter plot of observed versus ECHAM6.3–HAM2.3 simulated submicron sea salt concentration for the stations in Fig. 2. CVAO location has an additional identification as a reference to the measured marine compounds to facilitate the comparison with Fig. 7 (see also Table 2 and Table 4 for the compound and location abbreviations, respectively).

concentrations in the model.

### 6.2.3 Polar lipids

The scarce measurements of lipid aerosol within the simulated period, limit our comparison to only the location of CVAO
(Fig. 7 third panel). For this site, the measured concentrations and OMF exhibit significant disparities for $PG_{aer}$ when compared to $PCHO_{aer}$ and $DCAA_{aer}$. However, the model is capable of accurately representing the substantial enrichment of lipids in aerosols. Observations of $PG_{aer}$ are widely spread, with a large interquartile range. The OMF values range between $4.9 \times 10^{-4}$ and 0.15, whereas the aerosol concentration goes from $1.8 \times 10^{-3}\ \mu\mathrm{g\,m^{-3}}$ to $0.3\ \mu\mathrm{g\,m^{-3}}$. Conversely, simulated values do not present such variability.

Model OMF lays at the upper end of the OMF from measurements ($9.9 \times 10^{-2} \pm 1.4 \times 10^{-1}$) with a mean bias of 0.09. Similarly, the aerosol concentration is also overestimated (bias = $0.15\ \mu\mathrm{g\,m^{-3}}$) and the model median quantity is $0.29\ \mu\mathrm{g\,m^{-3}}$. $PL_{aer}$ group may encompass other subgroups rather than $PG_{aer}$. This could partially explain the overestimation of aerosol quantities by a factor of three. However, these pronounced discrepancies, as previously mentioned, are strongly influenced by the misrepresentation of sea salt at this tropical station (CVAO_PG in Fig 8 with bias=$1.3\ \mu\mathrm{g\,m^{-3}}$). Furthermore, in addition to





the sea-air transfer via bubble bursting, there are other processes that influence the lipid concentration in aerosols. For instance, the presence of bacteria in marine aerosol particles could lead to atmospheric lipid production and degradation (Triesch et al., 2021b).

### 6.2.4 Total organic matter

Finally, we examine the total marine aerosol concentration and OMF against observed $OM_{aer}$ concentrations and total organic
mass fraction in submicron aerosol (Fig. 7 fourth panel). This helps to understand the potential contribution of marine aerosols to total $OM_{aer}$. The observed aerosol quantities are highest for CVAO (Fig 7) with an OMF and $OM_{aer}$ concentration of $0.5 \pm 0.2$ and $0.4 \pm 0.13$ $\mu g\,m^{-3}$, respectively. On the other hand, NAO has smaller OMF ($0.32 \pm 0.15$) and concentration ($0.22 \pm 0.09$ $\mu g\,m^{-3}$) values. This is followed by WAP, wherein the smallest quantities are approximately one order of magnitude smaller compared to the other stations ($0.12 \pm 0.16$ and $0.07 \pm 0.08$ $\mu g\,m^{-3}$).

Note that the consideration introduced here to calculate the $OM_{aer}$ based on measured OC can be variable and is a source of error. After emission molecules suffer chemical transformations Nonetheless, OMF is in good agreement with the observations, although it is slightly smaller for NAO and CVAO. The OMF model estimation for CVAO is approximately 0.15. This low mass fraction agrees with the results from van Pinxteren et al. (2023), where marine organic compounds accounted for approximately 49 % of OC (identified OC fraction) while the other 51 % remained unknown. As in their study, we found that the lipids
group has the highest contributions to $OM_{aer}$. The modelled OMF of $PL_{aer}$ accounts for up to 84 % of total OMF for CVAO whereas that of $DCAA_{aer}$ and $PCHO_{aer}$ is less than the 13 % and 3 %, respectively. The total concentration of biomolecules in the model is $0.1 \pm 0.2$ $\mu g\,m^{-3}$, which is comparable to observed $OM_{aer}$. Surprisingly, even though the measurement data are close in time, the modelled aerosol concentrations vary within a large range. In this case, the overestimation can also be observed for sea salt, which is often underestimated by the ECHAM6.3–HAM2.3 model.

Similarly, total organic mass enrichment and the aerosol concentration for NAO cannot solely be explained by the biomolecules in our model (bias = -0.12 $\mu g\,m^{-3}$). The median total modelled OMF and total marine aerosol concentration are 0.25 and 0.06 $\mu g\,m^{-3}$, respectively. This implies that other organic aerosols that have been transported or secondary organic aerosols that have formed in the atmosphere are likely to constitute a significant portion of the observed organic matter and are not derived from marine sources.

Finally, for the WAP, we found that the modelled OMF is larger than the observations (bias = 0.07). It is likely that a notable $PL_{aer}$ OMF during bloom peak in the Southern Ocean in January could explain this. The potential large contribution from lipids is not necessarily representative of the actual $PL_{aer}$ enrichment at this southern station. Additionally, aerosol concentration is found at the minimum of observations, analogous to what we have previously discussed of sea salt misrepresentation for WAP station.

The results of our model demonstrate the significant sensitivity of marine organic emissions to sea salt emissions, owing to their linear correlation in calculating emission mass flux (Eq. (10)). Figure 8 illustrates the wide spread of the data with a large model bias (0.39 $\mu g\,m^{-3}$). We have observed that the model tends to underestimate the concentration of SS for both Northern and Southern stations, whereas for the tropical location (CVAO), simulations are much higher than the observed amounts.





These regional disparities may suggest that the dependence on SST holds significant influence on the emission flux of SS.
However, sensitivity studies removing the Sofiev et al. (2011) SST dependence from the Long et al. (2011) source function
(not shown) proved otherwise, which was similarly found by Barthel et al. (2019) and Tegen et al. (2019).

Despite the uncertainties in representing the organic fraction and aerosol concentration for point locations, the modelled values range in the same order of magnitude as the observations for most stations. Furthermore, our model can still capture the differential aerosol enrichment of the biomolecule groups also visible in the observations, in which $PL_{aer}$ predominates in the
aerosols with contributions of up to two orders of magnitude greater than $PCHO_{aer}$ and $DCAA_{aer}$.

## 6.3   Comparison of PMOA with in-situ aircraft measurements

To highlight the critical importance of evaluating the model's ability to represent PMOA in remote oceanic regions, where ground-based measurements are unfeasible or species-resolved quantities are unavailable, we compare the simulated PMOA
concentrations with organic aerosol mass from ATom aircraft observations. This comparative analysis emphasizes our model's performance, particularly when including PMOA (SPMOAon experiment), against the case where PMOA is not considered (SPMOAoff experiment). Figure 9 shows the statistical measures of the comparison of daily mass concentration of simulated PMOA in comparison to organic aerosols from ATom data. For the SPMOAoff case, we selected the total OC for the analysis, while the aggregated mass concentration of PMOA and OC was considered for the SPMOAon simulation. Note that modelled
OC only comprises emissions from anthropogenic sources and biomass burning.

Overall, the correlation with data increased for the SPMOAon experiment. The Pearson coefficient ranges from 0.35 for the SPMOAoff case to 0.45 for SPMOAon (not shown). Although, NMB and RMSE slightly increased for the latter. For most regions in Fig. 9, the Normalized Mean Bias (NMB) and Root Mean Squared Error (RMSE) tend to be low, except for the Northern Pacific and Atlantic areas, in which the statistical indicators could be up to 25-time larger. These two regions also
have the lowest correlation, and we will provide a more detailed analysis below.

By including PMOA, the representation of organic aerosols in the southern Pacific and Atlantic oceans is greatly improved. NMB is reduced nearly 88 % and 49 % for the South Pacific and Atlantic oceans, respectively. Similarly, correlation with observations is also higher for the SPMOAon experiment. It increases from 0.42 (0.65) for the SPMOAoff case to 0.48 (0.82) for SPMOAon in the South Pacific (Atlantic) ocean. The simulated aerosol concentration was underestimated for these re-
gions (Fig. B1(a, b)). Model values increased by one order of magnitude when PMOA is considered improving the estimation (Fig. B2(a, b)). Conversely, for the central Pacific, the Pearson coefficient shows little variation for both experiments (0.64). As for the southern regions, NMB is negative for the SPMOAoff case, underestimating the measured amounts (Fig. 9(a)). Initial OC concentrations for the region are within the range of observations (Fig. B1(c)). When PMOA is considered, model values rise by a factor of two (Fig. B2(c)). Consequently, NMB and RMSE augments by a factor of about 1.7 for the Central Pacific.
This is surprising, since the aerosol load in this region is expected to be governed by PMOA. We could speculate that the overestimation when adding PMOA is due to potentially too high sea salt mass in the accumulation mode or not efficient wet aerosol removal. However, given the lack of SS data in the sub-micron mode, it is not possible to provide conclusive arguments



in this regard.

Note that measured organic aerosols (OA) from ATom aircraft data encompass multiple aerosol sources and types. The OA
includes primary and secondary organic aerosols (Hodzic et al., 2020), which we were unable to disentangle from the original
dataset. In our model simulations, the primary aerosol was solely considered, and thus secondary aerosol formation resulting
from chemical transformation or oxidation is not included in the model. Nonetheless, our findings suggest that by adding
PMOA to primary non-marine OC, the model estimates are improved for southern oceanic regions.

Pai et al. (2020) also found a strong influence of primary marine organic aerosols in the lower levels of the atmosphere of
remote marine regions. Their model simulations tend to overestimate the observed values, with larger biases than in this study.
This might be the result of the differences in the marine aerosol emission parameterization (following Gantt et al., 2015) and
further transformation in the atmosphere.

On the other hand, the results for the Northern Hemisphere seem to be contradicting. By applying the criteria introduced in
Sect. 4.2 we limit the comparison to the marine boundary layer, in which marine local sources should predominate. However,
this is not always the case for the Northern Hemisphere, where anthropogenic continental sources dominate in the atmospheric
column (Pai et al., 2020). Furthermore, the weak correlations are strongly influenced by outliers in the simulated OC, resulting
from the influence of biomass burning and natural fires (Fig. B1(d, e)). Outliers correspond to peak values over 0.5 $\mu$g m,[3]
(values at standard conditions of temperature and pressure) indicating the pronounced influence of other sources rather than
marine (Fig. B1(d, e) and Fig. B2(d, e)). Moreover, for the northern regions, the vertical profiles of measured OA had a signifi-
cant occurrence in higher levels, between 300 and 1000 m (not shown). In contrast, for the central Pacific and southern oceans,
about 90 % of the data were confined to heights below 400 m. This indicates that probably transported aerosols constituted the
mass of OA in the higher marine boundary layer. In addition, we did not apply any threshold to the model data, as it would
result in a decrease in the sample size and a reduction in its representativeness. We believe that the aforementioned factors
could explain the poor correlation and large biases in the North Atlantic and Pacific oceans for both experiments.

Lastly, we found that for the North Pole, including PMOA also yields to worse correlations. The Pearson coefficient is reduced
by a factor of two for SPMOAon case, while RMSE and NMB show little variation. This suggests that the influence of marine
organic aerosols has a negligible impact in the high latitudes (Fig. B1(f) and Fig. B2(f)). Nonetheless, seven out of eleven
ATom samples were measured between October and April, a period in which PMOA concentrations are at their low given the
limiting nutrient and light conditions in the Arctic Ocean during the polar night. However, in most of the remaining samples
from summer, adding PMOA did not have any influence potentially related to measurements over the ice pack and the marginal
ice zone for which PMOA is less relevant.

Lastly, the model coarse temporal and spatial resolution is a major source of uncertainties for this comparison. The large daily
standard deviation in measured OA (Fig. B1 and Fig. B2) indicates the strong variability in the data, which, given our model
limitation, cannot be captured. Nonetheless, our results show that by including PMOA, our model represents better the organic
aerosol in the lower atmosphere of remote southern oceanic regions. Moreover, seasonal variability, which was not considered
here, seems to be a critical factor when validating aerosol model results with ATom data (Gao et al., 2022). Additionally, Gao
et al. (2022) also found better agreements when updates in the wet scavenging parameterization were incorporated into the





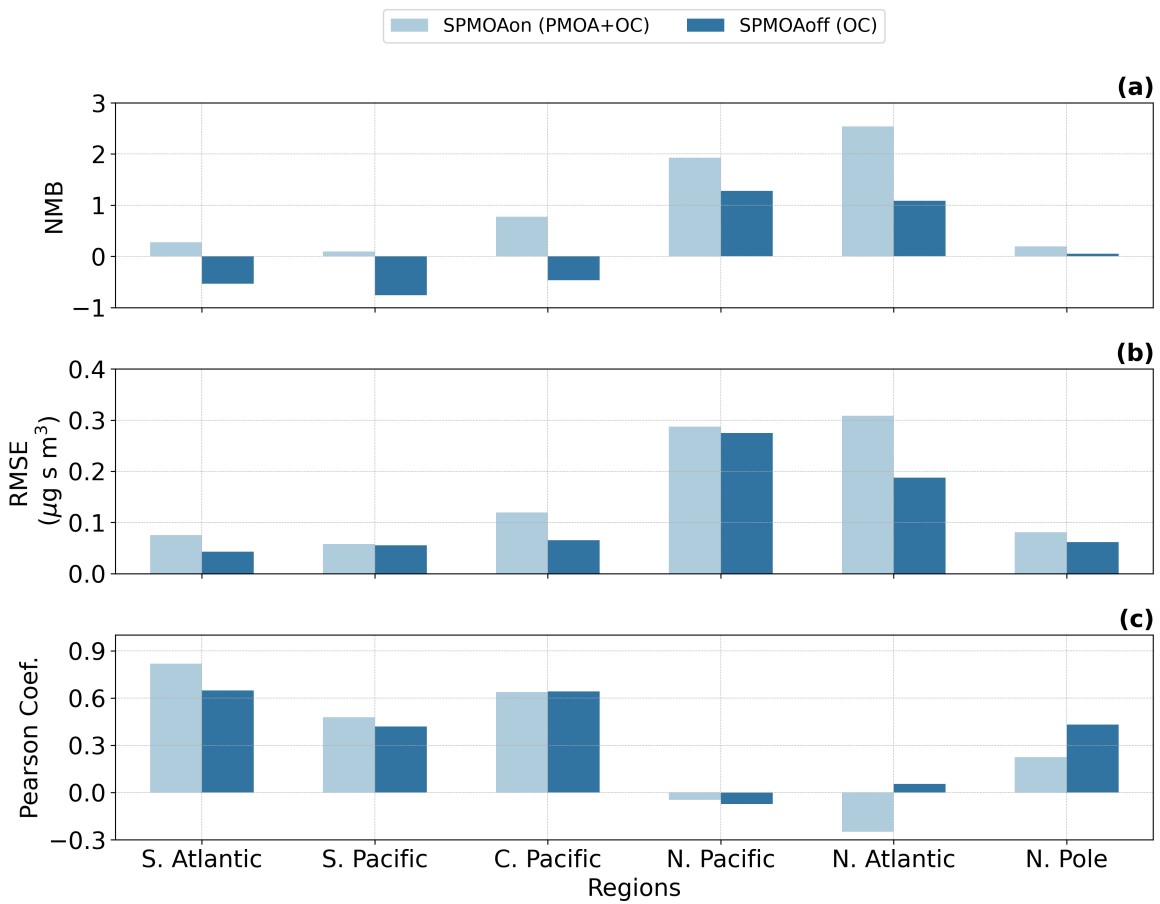

**Figure 9.** Bar plot of the (a) normalized mean bias (NMB), (b) root-mean-square error (RMSE) and (c) Pearson correlation coefficient (Pearson Coef.) of the comparison of daily averages of ECHAM6.3–HAM2.3 simulated PMOA and organic aerosol aircraft observations (ATom data). The data were grouped per region as in Fig. 3. Light and dark blue bars are the results from SPMOAon and SPMOAoff experiments, respectively. The formulas to calculate the statistical indexes may be found in Table A1.

model.

## 7 Summary and Conclusions

In this study, primary marine organic aerosols are included into the global aerosol-climate model ECHAM6.3–HAM2.3 to investigate their emission characteristics and distribution. The representation of organic mass fraction is based on the marine emission scheme OCEANFILMS by Burrows et al. (2014). We account for the contribution of three main marine biomolecule



groups in the ocean relevant for the aerosol phase (dissolved acidic polysaccharides (PCHO$_{sw}$), dissolved combined amino

acids (DCAA$_{sw}$) and polar lipids (PL$_{sw}$). Marine quantities are based on phytoplankton exuded carbon calculated by the

ocean biogeochemistry model FESOM2.1-REcoM3 and serve as bottom boundary condition to the atmospheric model.

The simulated PCHO$_{sw}$ is the most abundant group in seawater, followed by DCAA$_{sw}$ and PL$_{sw}$, both of which exhibit lower

contributions to the extracellular dissolved organic carbon. The abundance and amounts of molecules are well represented in

the simulations compared to worldwide in-situ seawater measurements.

The individual marine groups show a diverse global distribution. PL$_{sw}$ ocean concentrations are generally higher in biologi-

cally active regions, whereas PCHO$_{sw}$ and DCAA$_{sw}$ groups predominate in subtropical and tropical waters. The distribution

of the organic mass fraction of aerosols closely resembles that of the concentrations of biomolecules in oceans. Nevertheless,

a differential enrichment in the aerosols was detected. Given the high air-water surface affinity of lipids, this group dominates

the organic mass in aerosols while, PCHO$_{aer}$ and DCAA$_{aer}$ tend to be significantly lower in magnitude.

The seasonal patterns of the marine biomolecule and organic aerosol mass fraction were additionally examined. For polar

regions, seasonality is remarkable in contrast to mid- and equatorial latitudes. The greatest contribution of the three organic

groups occurs during summer in each hemisphere, and the values remain invariant for the Equator. In a follow-up study, we

will comprehensively examine the distinctive characteristics of the Arctic and the evolution of marine biomolecules and PMOA

over the past decades, owing to the distinct significance of this region and its prominent alterations in response to changing

climate conditions.

The global aerosol model simulations indicate that PMOA emission fluxes are sensitive to multiple factors. Among them, the

marine biological activity and surface wind conditions are mainly controlling the occurrence of PMOA. Due to elevated wind

speeds, the emission fluxes in the North Atlantic, North Pacific, and Indian waters tend to be stronger. The model estimate of

the aggregated PMOA emission globally is 13.6 Tg per year, corresponding to an atmospheric burden of 0.068 Tg.

Furthermore, the modelled mass concentrations of the primary marine organic aerosol was thoroughly evaluated using obser-

vations. A comparative analysis of PMOA against organic aerosols from NASA's ATom aircraft campaigns indicated that the

model has lower biases and better correlation for remote Southern Hemisphere oceans when PMOA is included. The species-

resolved aerosol evaluation showed that the model could capture the differential organic enrichment in aerosols. Nevertheless,

further improvements are needed to the aerosol-climate model. For instance, the underestimation of sea salt emissions affected

a more correct representation of the PMOA concentrations. Here, more detailed emission functions of sea salt or sensitivity

studies with multiple source functions could lead to better estimates in future (e.g., Grythe et al., 2014; Albert et al., 2016).

Additionally, adjusting OCEANFILMS parameters to better characterize the physicochemical properties of the biomolecules

studied here, could lead to a more accurate representation of the organic fraction in aerosols.

The different emission and transport patterns, found in this modelling study, suggest that the distinct components of PMOA

may exert a differentiated influence on cloud and precipitation formation. Our research provides a model setup that considers

diverse marine organic aerosol groups. This model setup will be further developed to incorporate the significance of indi-

vidual groups or aggregated PMOA in aerosol-cloud interaction processes in future studies. The capability of regional grid

refinements for the biogeochemical model provides a vast scope for the approach presented here to compute biomolecules as





boundary conditions for a high-resolution aerosol-cloud model simulation. This could be particularly interesting for Arctic studies, as it enables the incorporation of spatial fine structures, such as phytoplankton blooms along marginal ice zones, and evaluate the effect of mixed-phase clouds in the Arctic climate.

## Appendix A: Statistical index formulas

**Table A1.** Formulas of statistic indexes. $M_i$ and $O_i$ refers to the model results and observational data i for each station or region. N is the data sample size.

| Statistics | Formula |
|---|---|
| Mean bias | $\text{MB} = \frac{1}{N}\sum_{i=1}^{N}(M_i - O_i)$ |
| Normalize mean bias | $\text{NMB} = \frac{\sum_{i=1}^{N}(M_i - O_i)}{\sum_{i=1}^{N} O_i}$ |
| Root Mean Square Error | $\text{RMSE} = \sqrt{\frac{1}{N}\sum_{i=1}^{N}(M_i - O_i)^2}$ |





## Appendix B: Modelled aerosol concentration versus ATom aircraft observations

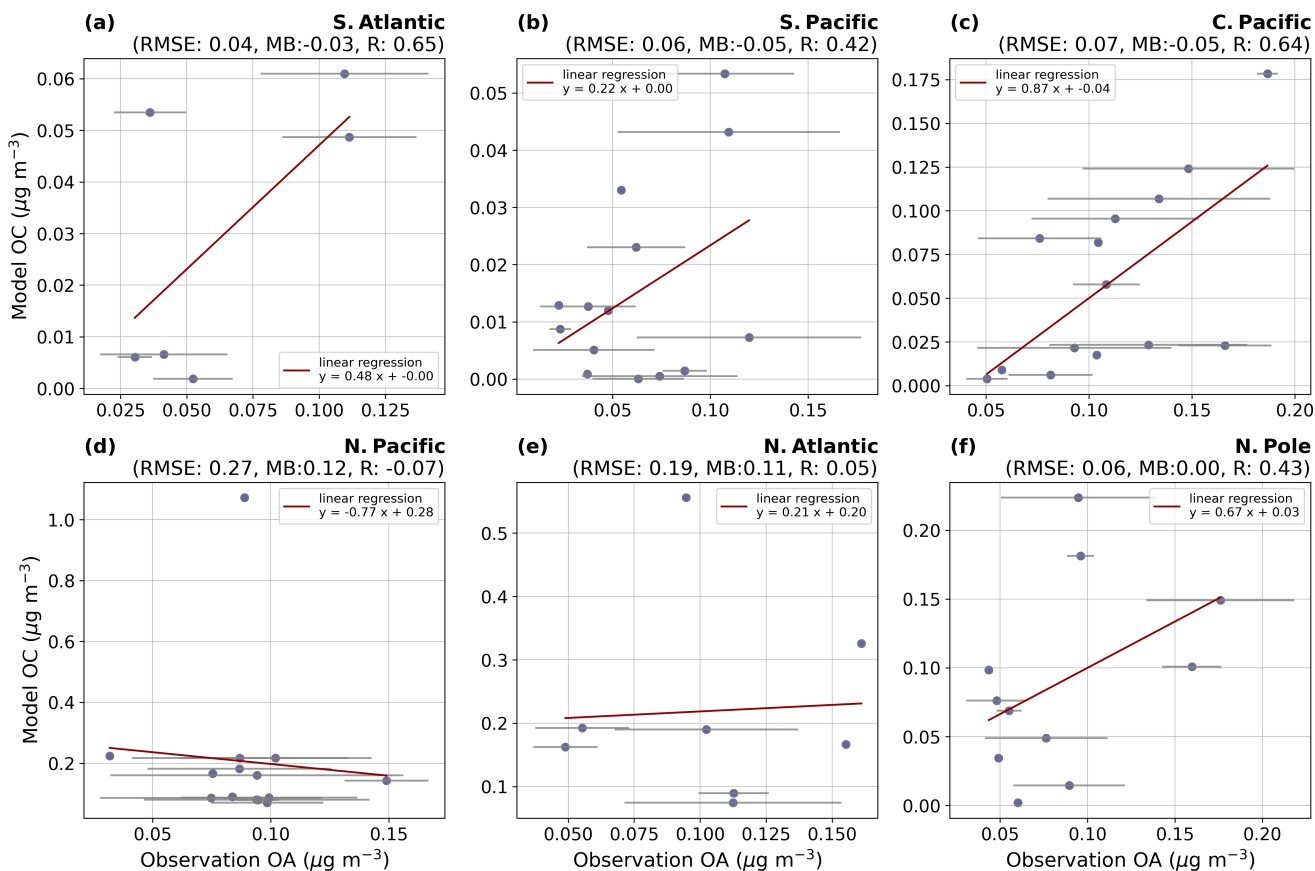

**Figure B1.** Scatter plot of the comparison of aerosol concentration of daily averages of ECHAM6.3–HAM2.3 model simulated PMOA and organic aerosols aircraft observations (ATom data) for the SPMOAoff experiment. The data were grouped per region as in Fig. 3. The red line indicates the linear regression fit. Light gray represents the daily standard deviation of observations. Observed and measured quantities represent the concentration at standard temperature and pressure conditions (273 K, 1 atm). The formulas to calculate the statistical indexes may be found in Table A1.



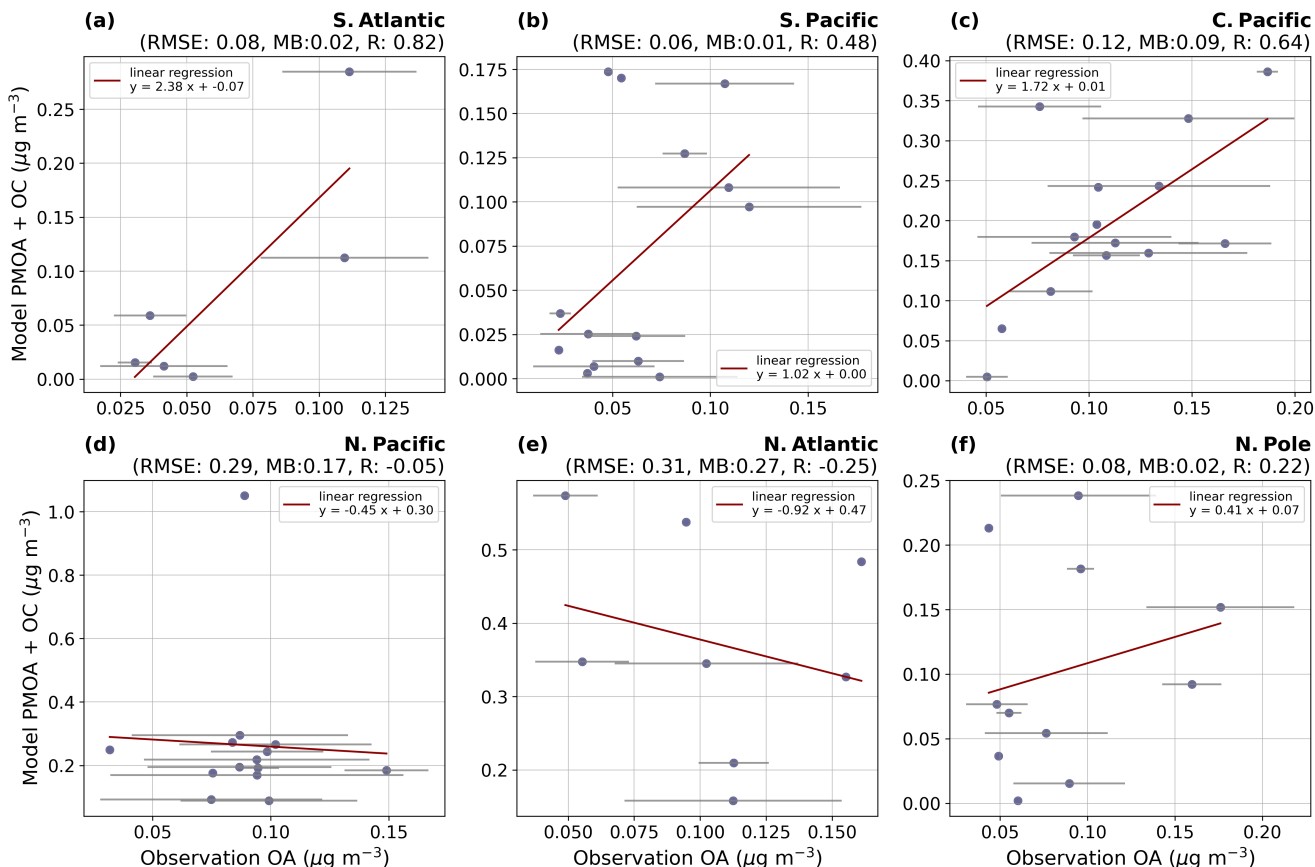

**Figure B2.** Scatter plot of the comparison of aerosol concentration daily averages of ECHAM6.3–HAM2.3 model simulated PMOA and organic aerosols aircraft observations (ATom data) for the SPMOAon experiment. The data were grouped per region as in Fig. 3. The red line indicates the linear regression fit. Light gray represents the daily standard deviation of observations. Observed and measured quantities represent the concentration at standard temperature and pressure conditions (273 K, 1 atm). The formulas to calculate the statistical indexes may be found in Table A1.



## Appendix C: Maps of monthly biomolecule quantities

**Figure C1.** Maps of monthly averaged ocean carbon concentration of PCHO$_{sw}$ (first column), DCAA$_{sw}$ (second column), and PL$_{sw}$ (third column) for February (a–c), May (d–f), August (g-j), November (k-m) as a multiannual mean for the period 1990-2019 for sea ice free conditions (SIC $<10\,\%$).





**Figure C2.** Maps of monthly averaged offline computed OMF of PCHO$_{aer}$ (first column), DCAA$_{aer}$ (second column), and PL$_{aer}$ (third column) for February (a–c), May (d–f), August (g-j), November (k-m) as a multiannual mean for the period 1990-2019 for sea ice free conditions (SIC $<10\,\%$).



*Code and data availability.* The source code of FESOM2.1-REcoM3 model is available at https://github.com/FESOM/fesom2. The biogeo-
chemistry model tracers to compute the marine biomolecules groups, the ocean biomolecule concentration and the results of the aerosol-
climate model simulations are accessible at https://zenodo.org/records/13235178. The ECHAM6.3–HAM2.3 code is managed and provided
through the website https://redmine.hammoz.ethz.ch/. Access to the code is governed by the terms set forth in the HAMMOZ Software
Licence Agreement. The seawater and aerosol measurement data sets were derived from the referenced literature or requested directly to
the authors. The in-situ aircraft observations were acquired as a data collection shared at The Oak Ridge National Laboratory Distributed
Active Archive Center (ORNL DAAC) https://doi.org/10.3334/ornldaac/1925. The model evaluation with observations was performed with
python3 (Python Software Foundation) with available interpolation functions in the module SciPy. In addition, to handle the model data and
visualize the results, other python3 libraries such as xarray, pandas, cartopy, and matplotlib were used. Climate Data Operators (cdo) version
2.2.4 were used for adapting the bottom boundary condition data to the ECHAM6.3–HAM2.3 grid and for the posterior interpolation of the
aerosol model results to a regular vertical and horizontal grid.

*Author contributions.* AL-M developed the approach to compute the biomolecules based on FESOM2.1-REcoM3 model tracers and im-
plemented the marine organic aerosol parameterization into the aerosol model; MZ incorporated the PCHO and TEP as tracers in the bio-
geochemistry model; MZ and AL-M did the postprocessing of the ocean data output; AL-M and BH planned the aerosol-climate model
simulations; AL-M performed the models evaluation; MvP, SZ and BH provided scientific advice in the process understanding and model
representation of organics abundance in seawater and in aerosol; MvP, SZ, AB, EB, AE and MF facilitated the access and assisted with most
of the data employed for the models evaluation; MZ wrote the biogeochemistry model description; AL-M wrote the remaining sections of
the manuscript draft; MZ, MvP, SZ, AB, EB, AE, MF, IT and BH reviewed and edited the manuscript; BH supervised and supported the
work by providing scientific feedback to all sections; AB, BH and MvP acquired funding.

*Competing interests.* The authors declare that they have no conflict of interest.

*Acknowledgements.* We gratefully acknowledge the funding by the Deutsche Forschungsgemeinschaft (DFG; German Research Foundation;
project number 268020496; TRR 172) within the Transregional Collaborative Research Centre "ArctiC Amplification: Climate Relevant At-
mospheric and SurfaCe Processes, and Feedback Mechanisms $(AC)^3$". We are also grateful for the active and collaborative work among the
subprojects B04, C03 and D02's members within $(AC)^3$. We especially thank the developers of ECHAM-HAM. The ECHAM-HAMMOZ
model is developed by a consortium composed of ETH Zürich, Max Planck Institute for Meteorology, Forschungszentrum Jülich, the Uni-
versity of Oxford, the Finnish Meteorological Institute and the Leibniz Institute for Tropospheric Research and managed by the Centre for
Climate Systems modelling (C2SM) at ETH Zürich. The authors are grateful for computing time from the Deutsches Klimarechenzentrum
(DKRZ). Computing resources at DKRZ were granted under project number bb1005. The authors express their gratitude for the computing
time granted by the Resource Allocation Board and utilized on the supercomputers Lise and Emmy at NHR@ZIB and NHR@Göttingen
as a component of the NHR infrastructure. The calculations for this research were conducted with computing resources under the project
hbk00084. We are thankful to Dr. Susannah Burrows (Scientist at Pacific Northwest National Laboratory) for the assistance provided in



the offline implementation of OCEANFILMS and facilitating the ocean macromolecules data from CESM biogeochemical modules for test runs. We appreciate Swetlana Paul's (PhD candidate at TROPOS) assistance in the implementation and improvement of some post-processing python tools employed for the model evaluation.



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
