# Peer review of "Modelling emission and transport of key components of primary marine organic aerosol using the global aerosol-climate model ECHAM6.3–HAM2.3"

_EGUsphere, 2024_

## Author Comment (AC5)

**Response to RC1: 'Comment on egusphere-2024-2917'**

February 6, 2025

We would like to thank the reviewer for revising our manuscript and for the valuable feedback you provided. Your insightful comments and constructive suggestions have helped in improving the clarity and quality of our work. We truly appreciate the careful attention you paid to both the strengths and areas for improvement within our study. We made every effort to address all the comments you provided in your review. Please find below the response (in blue) to your comments (in black).

General comment

The will of the authors to be very thorough is good, but the manuscript is very long and dense as a consequence. I recommend the authors try to trim down the manuscript by removing parts that are non-essential. As an example, the sentences line 265-266 are simply a repletion/reformulation of the previous sentence line 264-265, while that first sentence contains already all the information needed on how the module represents aerosols. There are several such examples throughout the manuscript and I trust the authors can reduce the manuscript length without losing any information by removing statements that would be obvious to GMD readers.

We followed this helpful suggestion and rephrased and merged multiple sentences throughout the text, including the instances highlighted by the reviewer.

Minor comments

1. The wording in the Abstract is sometimes too strong, e.g. 1.1. "The comparison shows a strong agreement, given the uncertainties in model assumptions and measurements." - Seeing your results, I would not call this a strong agreement, especially in the Northern Hemisphere compared to Atom.
   1.2. "model biases in the representation of the marine organic aerosol groups are caused by uncertainties in the simulated sea salt concentrations" - as mentioned below in another comment, I do not think you can attribute the model biases solely to sea salt aerosol based on what you show.
   We see your point and agree. We changed the wording as follow: 1.1. "The comparison shows a reasonably good agreement, given the uncertainties in model assumptions and measurements."
   1.2. "Model biases in the representation of the marine organic aerosol groups are caused by uncertainties in the aerosol-process representation and the simulated sea salt concentrations."
   See also response to comments 4, 5 and 6.

2. Section 3.3 - It is unclear why you cannot simply use the SIC and SST from FESOM-REcoM to force ECHAM-HAM. Instead you use this "SIC and SST mask", where you sometimes use AMIP data and sometimes replace it with FESOM-RECoM, that I do not fully understand. What is the reason for not using SIC and SST from FESOM-REcoM always?

   We agree that using the same SIC and SST would have been the more consistent approach. However, tuning the model against new boundary conditions is a complex and lengthy process that was beyond the scope of this study. Therefore, we kept the AMIP SST and SIC in order to avoid alterations of the atmospheric dynamics in the current model version, which has been extensively evaluated in previous studies. As a compromise, given the importance of the open ocean for marine emissions, we decided to utilize the fine resolution of the FESOM grid by refining the AMIP

SIC and SST in the marginal ice zone using the SIC values from FESOM-REcoM SIC for the emission scheme. Note that we do not consider the actual SIC values from FESOM-REcoM for the calculation. For the marine emissions, we create a mask based on the ocean model data that removes grid cells covered by sea ice from the AMIP grid when FESOM SIC fulfils the criteria for open ocean conditions ($SIC < 10\%$). This results in emissions not being limited by a higher AMIP SIC in grid cells, where marine organic aerosol production could potentially occur on a larger scale given the low SIC in FESOM. Nevertheless, only a small fraction of AMIP grid cells is actually affected by the mask, and comparison with observations confirms our approach.

We included this in lines 303-307: "... SPMOAoff and SPMOAon respectively. The SPMOAoff simulation only accounts for the fraction of sea salt in sea spray aerosol, whereas the SPMOAon utilizes the biomolecule ocean surface concentration as bottom boundary conditions to compute the marine organic aerosol fraction in addition to the sea salt. For consistency with the biogeochemistry model prognosticated sea ice, an adjustment of SIC and SST within the sea salt emission scheme is considered, intending to avoid ambiguities. To ensure comparability with previously published results of the aerosol-climate model (Tegen et al., 2019) and avoid re-tuning, the AMIP data is retained for the simulations. A mask is applied to determine when FESOM2.1-REcoM3 model SIC and SST values replace or modify AMIP data. Whenever ice free..."

3. L.519-527: here is another example of text that could be taken out. This paragraph is solely a description of the values in the measurements, without reference to modelled values. Although this bit of analysis is not uninteresting, I think it is unnecessary and contributes to making the manuscript too dense.
Thank you for the suggestion. We followed your recommendation and reduced/merged some paragraphs throughout the manuscript. Changes are highlighted in the edited version of the manuscript.

4. L.584-587: According to Equation 10, the ratio of PMOA to SS should be the sum of OMFi/(1-OMFi), right? But then OMFi does not depend on the sea salt source function according to the formulas provided. Therefore I am not sure your explanation is actually valid, unless I missed something.
In this section by ratio of PMOA to SS, we did not mean to refer to the OMF but rather the ratio of the aerosol emission mass fluxes ($\frac{\text{PMOA}_{massflux}}{\text{SS}_{massflux}}$). Thanks to this comment, we realize this statement could be confusing. Hence, we refer to this ratio as: "... the ratio of PMOA to SS emission mass fluxes..." (see also response to the following question)

5. L.588-593: I do not follow the argument here. What is the basis for saying that when you compare a chl-a based OMF and yours the difference in the results is still more driven by the sea salt than by the fundamental differences in modeling OMF?
We strongly agree with the reviewer here and modified the text from line 585-593 : "... from 0.048 to 0.097 Tg (Huang et al., 2018; Zhao et al., 2021; Burrows et al., 2022). Given the similarities in terms of model configuration, our values are closer to the results by Huang et al. (2018), despite the chl-$a$ approach used in their study to compute PMOA. This indicates that the driving aerosol-climate model has a greater influence on the final computed PMOA emissions than the specific representation of marine organics. There is also likely attributable to the sea salt emission scheme employed within the model. Therefore, the ratio of PMOA to SS emission mass fluxes varies across studies and in our case, it is larger ($1\%$, Table 8) than the ratios presented by Zhao et al. (2021)($0.67\%$) and Meskhidze et al. (2011) ($0.7\%$). "

6. Figure 6: why do you mask out land on the total burden panel (b)? It would be very interesting to see how far inland PMOA can be transported in your model and evaluate if they can affect atmospheric composition and clouds over continental areas.
We agree with this suggestion and removed the land mask in the maps.
In the process of reviewing your suggestions, we stumbled with a bug in the model related to

a missing factor in the computation of the PMOA number mass flux. This was immediately solved and tested. The updated results differ from the values previously presented in Table 8 and in Fig. 6. However, the model evaluation analysis remains nearly identical, as the changes occurred within the range of the model standard deviation for most cases. All figures affected by this were updated in the current manuscript version, with only small changes compared to the former version.

7. L.632-637: here is another unnecessary paragraph. The limitations of comparing coarse resolution models to observations are well-known and do not need to be explained in a GMD article.
We understand and agree that these explanations are not necessary for an GMD article, so we have removed them.

8. Section 6.2.1: here too you could merge the first 2 paragraphs to improve readability.
See response to comment number 3

9. L.744-746: SST is usually a 2nd-order driver of sea salt emissions, compared to the more important wind speed. I would take this discussion out, as surely the regional differences can be better explained by the ability of the model to represent winds in different regions.
We agree in this regard and this paragraph was removed.

10. L.801-802: could you illustrate this statement with references? I think it is a little bold to suggest that PMOA do not matter in the Arctic solely based on your comparison with observations. Biases in your simulation could also explain the loss of correlation, like transport and removal, or the oceanic biogeochemistry in the Arctic Ocean that can be challenging to represent. Please moderate the message here or provide references that support this claim.
We did not intend to diminish the importance of marine organic aerosols in the Arctic. Aerosol-climate models are generally strongly challenged in the polar region (Sand et al., 2017; Schmale et al., 2021; Whaley et al., 2022), with limitations in processes representation. Model biases in the simulation of ocean biomolecules, aerosol transport, transformation, and removal in the Arctic region could influence the accuracy of the modelled quantities.
However, following the recommendation from RC2 (point 16), we decided to exclude the flight trajectories of the ATom dataset in the Northern Hemisphere, where aerosols are often influenced by anthropogenic sources, biomass burning, or natural fires. This exclusion helps prevent weak PMOA signals from being masked by more dominant aerosol contributions. This is also valid for the Arctic, where more than half of the dataset was only available from October to April. During this period, local marine organic contributions from the open ocean are minimal and thus overshadowed by long-range-transported aerosols from lower latitudes.

11. Figures C1-C2: the figures would be easier to read if instead of letters for panel titles you put directly the month of the data as the title of the panel.
We considered this suggestion and updated the letters to the respective month in Figures C1-C2.

**References**

Burrows, S. M., Easter, R. C., Liu, X., Ma, P. L., Wang, H., Elliott, S. M., Singh, B., Zhang, K., and Rasch, P. J.: OCEANFILMS (Organic Compounds from Ecosystems to Aerosols: Natural Films and Interfaces via Langmuir Molecular Surfactants) sea spray organic aerosol emissions - implementation in a global climate model and impacts on clouds, Atmospheric Chemistry and Physics, 22, 5223–5251, https://doi.org/10.5194/acp-22-5223-2022, 2022.

Huang, W. T. K., Ickes, L., Tegen, I., Rinaldi, M., Ceburnis, D., and Lohmann, U.: Global relevance of marine organic aerosol as ice nucleating particles, Atmospheric Chemistry and Physics, 18, 11 423–11 445, https://doi.org/10.5194/acp-18-11423-2018, 2018.

Meskhidze, N., Xu, J., Gantt, B., Zhang, Y., Nenes, A., Ghan, S. J., Liu, X., Easter, R., and Zaveri, R.: Global distribution and climate forcing of marine organic aerosol: 1. Model improvements and evaluation, Atmospheric Chemistry and Physics, 11, 11 689–11 705, https://doi.org/10.5194/acp-11-11689-2011, 2011.

Sand, M., Samset, B. H., Balkanski, Y., Bauer, S., Bellouin, N., Berntsen, T. K., Bian, H., Chin, M., Diehl, T., Easter, R., Ghan, S. J., Iversen, T., Kirkevåg, A., Lamarque, J.-F., Lin, G., Liu, X., Luo, G., Myhre, G., van Noije, T., Penner, J. E., Schulz, M., Øyvind Seland, Skeie, R. B., Stier, P., Takemura, T., Tsigaridis, K., Yu, F., Zhang, K., and Zhang, H.: Aerosols at the poles: an AeroCom Phase II multi-model evaluation, Atmospheric Chemistry and Physics, 17, 12 197–12 218, https://doi.org/10.5194/acp-17-12197-2017, 2017.

Schmale, J., Zieger, P., and Ekman, A. M. L.: Aerosols in current and future Arctic climate, Nature Climate Change, 11, 95–105, https://doi.org/10.1038/s41558-020-00969-5, 2021.

Tegen, I., Neubauer, D., Ferrachat, S., Drian, C. S.-L., Bey, I., Schutgens, N., Stier, P., Watson-Parris, D., Stanelle, T., Schmidt, H., Rast, S., Kokkola, H., Schultz, M., Schroeder, S., Daskalakis, N., Barthel, S., Heinold, B., and Lohmann, U.: The global aerosol–climate model ECHAM6.3–HAM2.3 – Part 1: Aerosol evaluation, Geoscientific Model Development, 12, 1643–1677, https://doi.org/10.5194/gmd-12-1643-2019, 2019.

Whaley, C. H., Mahmood, R., von Salzen, K., Winter, B., Eckhardt, S., Arnold, S., Beagley, S., Becagli, S., Chien, R.-Y., Christensen, J., Damani, S. M., Dong, X., Eleftheriadis, K., Evangeliou, N., Faluvegi, G., Flanner, M., Fu, J. S., Gauss, M., Giardi, F., Gong, W., Hjorth, J. L., Huang, L., Im, U., Kanaya, Y., Krishnan, S., Klimont, Z., Kühn, T., Langner, J., Law, K. S., Marelle, L., Massling, A., Olivié, D., Onishi, T., Oshima, N., Peng, Y., Plummer, D. A., Popovicheva, O., Pozzoli, L., Raut, J.-C., Sand, M., Saunders, L. N., Schmale, J., Sharma, S., Skeie, R. B., Skov, H., Taketani, F., Thomas, M. A., Traversi, R., Tsigaridis, K., Tsyro, S., Turnock, S., Vitale, V., Walker, K. A., Wang, M., Watson-Parris, D., and Weiss-Gibbons, T.: Model evaluation of short-lived climate forcers for the Arctic Monitoring and Assessment Programme: a multi-species, multi-model study, Atmospheric Chemistry and Physics, 22, 5775–5828, https://doi.org/10.5194/acp-22-5775-2022, 2022.

Zhao, X., Liu, X., Burrows, S. M., and Shi, Y.: Effects of marine organic aerosols as sources of immersion-mode ice-nucleating particles on high-latitude mixed-phase clouds, Atmospheric Chemistry and Physics, 21, 2305–2327, https://doi.org/10.5194/acp-21-2305-2021, 2021.

---

## Author Comment (AC6)

**Response to RC2: 'Review of the manuscript by Leon-Marcos et al.'**

February 6, 2025

We appreciate the reviewer's thorough evaluation of our manuscript and the invaluable feedback provided. Your perceptive observations and constructive recommendations have significantly enhanced the clarity and quality of our work. We are grateful for the detailed attention you devoted to pinpointing the areas in need of improvement and clarity in our study. In response, we have addressed each of the points raised in your review. Please find below the response (in blue) to your comments (in black).

The modelling study by Leon-Marcos et al. addresses very important topic of primary marine organic matter in global ocean and atmosphere by incorporating the most advanced and further updated schemes into the global model. The paper is very well written and easy to follow, however, still requires some organisation to have more logical flow. The most disappointing aspect of the study, and not of the authors effort, is very poor representation of sea salt in sea spray. That does not undermine the authors efforts, but places low credibility on quantitative aspect of the results. If there is no agreement on the most basic species of submicron sea salt, how could we trust aerosol PMOA mass concentrations? Overall, the paper is a very significant contribution and representation of the state-of-the-art development of PMOA emission schemes and I, therefore, recommend publication of the study, but with a pinch of salt.

The major organisational comment relates to the sea salt comparison. It appears in various places and when suites the authors narrative, despite being the key species and the most basic parameter of the modelling effort, because PMOA is emitted by sea spray source function. BTW is it unclear what sea salt source function is used in the model in the first place. Whatever it was it should probably be avoided in the future? I suggest consolidating all sea salt comparison and information at the start of chapter 6.2 before discussing individual PMOA species.

The reviewer pointed out a crucial aspect in our work, and we appreciate the opportunity to justify our assumptions. The current standard configuration of ECHAM6.3-HAM2.3 used in the present study to represent sea salt follows Long et al. (2011) with a sea surface temperature correction by Sofiev et al. (2011) (please see section 3.2). We understand the limitations of the poor representation of SS for some stations; however, the source function considered here is the best possible representation of SS aerosols worldwide for the current ECHAM-HAM model version. Hence, we had to rely on the standard model configuration. A comprehensive comparison by Tegen et al. (2019) shows the range of several SS source functions, concluding that the scheme used here showed the most reasonable agreement of SS seasonality and aerosol concentration compared to observations. Additionally, for the stations analysed in our study, some sensitivity experiments with other implemented source function schemes in the model (not shown), confirmed that the standard setup leads to better results. We agree that the SS comparison to measurements should be provided in section 6.2, independent of the PMOA individual species. Therefore, a new subsection (6.2.1) has been included to present and discuss in more detail the concentration of SS submicron aerosols compared to measurements.

Other comments as they appeared:

1. Line 81. This paragraph ideally suites to start the follow up section.
   Since, with this paragraph, we summarize the content presented in each section of the manuscript, we believe it aligns better to the introduction than to Section 2. Nevertheless, thanks to your comment, we noticed that the reference to Fig. 1 did not suit the introduction. Hence, following your recommendation, we shift it to Section 2. Additionally, we introduce this section with a brief

description: "This section outlines the method used to determine the organic aerosol mass fraction and quantify biomolecule concentrations in the ocean, which is used in the aerosol-climate model simulations to account for species-resolved PMOA."

2. Equation 3. notation g m-3
The notation of Equation 3 is g m-2, since $M_{SS}$ is referring to the mass of sea salt per unit of area. Thanks to the reviewer comment, we realised that the description of the variables in lines 105-106 was not clear and led to confusions. We rewrote the sentence for the sake of clarity: "The organic mass fraction $(OMF_i)$ is then calculated based on the mass per bubble surface area of the individual macromolecule group $(M_i)$ and of sea salt $(M_{SS})$"

3. Line 159. Line 299 refers to 2009-2019 period. Clarification or correction for consistency needed.
We noticed that this information is confusing and needs clarification. The available biogeo-chemical model output spans from 1990 to 2019. However, the aerosol model simulations were performed only for a 10-year period (2009-2019) in this paper, prioritizing the years in which aerosol observations were available. For consistency, we modified line 299 as follows: "The simulations cover a period of ten years (2009-2019) in which aerosol measurements were available. A spin-up time of four months and an output frequency of 12 hours was considered." In a follow-up paper, which will be summited soon, the entire time span will also be analysed for the aerosol climate simulations with respect to Arctic trends and patterns.

4. Line 262. What SS source function is used in the model?
This was probably overlooked, but the SS source function and respective temperature correction considered in the model experiments are mentioned in lines 286-287. Nonetheless, for clarity, we extended the description in this section as follows: "... $SS_{massflux}$ is the mass flux of sea salt emitted in ECHAM6.3–HAM2.3. The model includes a range of widely used sea salt emission schemes, including, for instance Guelle et al. (2001), Gong (2003) and Long et al. (2011). The default configuration is considered for the current study and follows Long et al. (2011) with sea surface temperature correction according to Sofiev et al. (2011). This combination the best agreement with observations in the model evaluation study by Tegen et al. (2019).".

5. Line 269. Soluble and insoluble aerosol particles are introduced without the substance what biomolecule species would be contributing to each. Or not contributing?
We considered that all biomolecule groups are emitted as soluble particles. As it remained unclear what biomolecule would be contributing to each mode, Table 3 was modified, replacing "PMOA" by the biomolecule species abbreviations ($PCHO_{aer}$, $DCAA_{aer}$ and $PL_{aer}$). We clarified some critical aspects and reordered the text for the description of how PMOA are considered in the model: Now line 272: "...The PMOA compounds are treated as separate tracers and included in the model as three new individual species (Atmosphere compartment in Fig. 1) to the soluble accumulation and coarse modes (see Table 3). These organic groups do not contribute to but share the microphysical and optical particle properties of the OC tracer..."

Many studies consider lipids as insoluble material, although OCEANFILMS treats all biomolecules as dissolved organic carbon species. More detailed description is needed here, considering the above.
Surface-active compounds such as lipids can exist as dissolved in seawater. However, when aerosol samples collected on filters and analysed in the laboratory, the aerosols in the process of dissolving them again does not mirror the ocean conditions in which they originally existed as dissolved. As a result, these particles can appear as water-insoluble organic aerosol in atmospheric measurements, despite being "dissolved" or at least well-mixed in the ocean Frossard et al. (2014).
From the modelling perspective, OCEANFILMS does not account for higher enrichment of particulate material compared to the dissolved phase in the surface microlayer (SML) and bubble film Burrows et al. (2014). Hence, only the seawater dissolved organic fraction is considered.

Nonetheless, a study by Burrows et al. (2022) evaluated the model implementation of primary marine organic aerosols emitted as insoluble particles, which are eventually transferred to the soluble mode as particles age (as an external mixture with SS) and as soluble particles emitted together with SS (internally mixed with SS). The results presented in their study conclude that the fully internally mixed experiment led to better model estimations of the aerosol concentration and seasonal cycle, as well as their evaluated effect on clouds. Therefore, in our study, we also considered that PMOA is emitted solely in the soluble mode.

Later in the text it mentioned OC from other than marine sources, so perhaps insoluble material is entirely anthropogenic. However, it confusing to say the least.
Note that OC and PMOA as computed here are different independent model tracers. In the model, there are no natural marine sources contributing to the tracer OC. The origin of OC is either from anthropogenic sources or biomass burning. In the HAM module, OC is the only organic specie emitted as insoluble particle in the Aitken mode, whereas larger modes are only soluble.

6. Line 279. Since HAM is using M7 aerosol model which does not consider Aitken sea spray the study is missing out on recent observational evidence of multiple submicron sea spray modes. Aitken sea spray with OM may not contribute significantly to the fine particle mass fraction, but it can have profound effect on cloud activation. It worth mentioning here that the modelling study utilised the very state-of-the-art of PMOA in the ocean, but not state-of-the-art of sea spray generation, including temperature-flux relationship.
We agree that PMOA in the Aitken mode could have a great impact on clouds specially, due to their contribution in number mixing ratio, potentially favouring the INP population. Nevertheless, our model does not represent sea spray in the Aitken mode. This could be considered in future research when the INP capabilities of PMOA are evaluated. However, this topic is out of the scope of the current study. Instead, we would like to stress that the sea salt flux generation in the model actually does include wind- and temperature-dependency, following Long et al. (2011) and Sofiev et al. (2011). This combination corresponds to the most accurate and up to date SS representation for the current ECHAM-HAM model version, as shown by Tegen et al. (2019).

7. Line 302. Does that mean that SPMOAoff is only accounting for sea salt in sea spray?
This means that the SPMOAoff experiment simulates only a fraction of SSA that corresponds to SS. To clarify, we expand the description of the two simulations by: "The SPMOAoff simulation only accounts for the fraction of sea salt in sea spray aerosol, whereas the SPMOAon utilizes the biomolecule ocean surface concentration as bottom boundary conditions to compute the marine organic aerosol fraction in addition to the sea salt."

8. Line 320. There has to be a consideration what impact proximity of the stations to the continents or islands would have in comparison to modelled open ocean.
We found this comment very well aligns to question number 10 and provided the answer to it below.

9. Line 355. This is consistent with the results of the O'Dowd et al. 2015 in Scientific Reports study where correlation between Chl-a and OMF was degrading as the time resolution increased.
Thank you for the suggestion. We agree this might also be potentially true in our case. Nevertheless, Line 355, only refers to the evaluation with seawater samples, in which the comparison to monthly values of biomolecule concentrations indicated a better agreement with observations in contrast to daily values. Based on this premise, we could expect a better representation of OMF by using a coarser temporal resolution. However, we did not perform an analysis of how the model resolution could affect the correlation between the ocean biomolecules and OMF, or the aerosol representation.

10. Line 361. How is continental outflow treated in the model? It can be significant, especially closer to the land masses.

In the model, the continental outflow can be composed of OC from anthropogenic emissions and biomass burning, as well as recirculated PMOA. Land-based biogenic aerosol sources are not included. However, since OC and PMOA are separate tracers, the origin of the air masses can be easily distinguished.

Interpolated grid-averaged data to point measurements, as in other models, inevitably leads to uncertainties in the results. This especially applies to areas with inhomogeneous surfaces, such as coastlines, where air masses change from ocean to land and the aerosol field homogeneity within the grid cell may lead to errors. To address this matter, we selected an interpolation method that detects and accounts for gradients around the station location.

Do I understand correctly that the model completely neglects anthropogenic OM as PMOA comes directly from OCEANFILMS and sea spray emissions? However, filter samples may have captured continental outflow, unless sector specific sampling method has been performed. If BC was modelled, its presence would indicate the amount of continental impact which would ease comparison.

That is not correct. Anthropogenic organic matter is not neglected in the model. The ECHAM-HAM model includes the tracer OC for organic aerosols from anthropogenic or biomass burning sources. In addition and independent, there is the PMOA tracer (or its subspecies) that is composed of the marine organic fraction of sea spray aerosol. Therefore, in the model, the total organic matter is the sum of OC and PMOA. Biogenic emission from vegetation on land and secondary organic aerosol (SOA) formation, however, are not considered in this model configuration.

According to several criteria for air masses of marine origin as explained in Pinxteren et al. (2020), van Pinxteren et al. (2023), Triesch et al. (2021a), Triesch et al. (2021b) and because of the application of sector-specific sampling, Zeppenfeld et al. (2021, 2023), the aerosol samples were considered to be mainly of marine origin. Nevertheless, OC samples are not completely excepted from the contribution from long-range transported aerosol. Hence, following your comment (and question 16) we incorporated and discussed in now section 6.2.5., the results of the comparison of modelled PMOA+OC against observed OM (OM = OC(obs) multiplied by the conversion factor 2).

11. Line 594. Wind is indeed the driving factor behind sea spray emissions (mind Reynolds number as a more appropriate predictor), however, the flux parametrization is the key with the magnitude of the power law controlling sea spray mass at a more significant wind speed.

We agree that the exponential dependence of the emission flux on surface wind is indeed the driving factor of sea spray emissions. We removed the sentence in line 594 as it was contradicting this statement.

12. Line 605. What latitude Southern Ocean is limited to? Where roaring forties and screaming fifties belong?

The Southern Ocean was limited between 60 and 90 degrees South (as defined in Table 6).

13. Section 6.2 Section should start by evaluating sea salt in aerosol phase without simulated PMOA, because sea spray emission schemes are vital for correct prognosis of PMOA. Without such a comparison, PMOA simulation results are very academic: developed a scheme, ran the model and whatever comes out of it compares with measurements were differences somewhat speculatively explained.

Line 673. As commented earlier, comparison of sea salt should be comprehensively described at the very start, instead of in places where argument is needed.

We strongly agree with the reviewer that the sea salt evaluation should lead at the beginning of section 6.2. We added a new subsection and discussed in detailed the SS model representation compared to observations (section 6.2.1). Also note that, we added a box-plot-style diagram to represent sea salt (Figure 7), for similarity and better reference to the figure of the species-resolved

analysis. A more comprehensive evaluation is not presented in the current study, as a thorough analysis and sensitivity studies of SS emission were already performed by Tegen et al. (2019). Based on the evaluation of the SS scheme in Tegen et al. (2019) and with full awareness of the uncertainties and potential underestimations in specific regions, we are confident that our PMOA results are scientifically robust and suitable for meaningful comparison with aerosol observations.

14. Figure 8. This is rather depressing comparison, because there is no agreement between model and observations of a key sea salt. It does not invalidate the simulation effort of PMOA using sophisticated schemes, but rather places very low credibility of modelled PMOA.

Fig. 8 indeed shows a substantial underestimation of modelled SS, especially in the Southern Hemisphere, while SS around Cape Verde is overestimated by the model. Here the agreement may be poor but not entirely absent. We agree that having a more accurate SS representation would determine a higher credibility in our quantitative results. However, evaluating which source function would be the most suitable is beyond the scope of the current study. And the question is probably difficult to answer, which shows the wide variety of SS emission schemes that have been extensively evaluated (Grythe et al., 2014; Barthel et al., 2019; Lapere et al., 2023). Large uncertainties remain in the SS representation when an empirical function based on local field or laboratory measurements is applied to the variety of atmospheric conditions around the globe. Barthel et al. (2019) and Lapere et al. (2023) are recent studies showing that the large heterogeneity in SS distribution and abundance, among other factors, strongly depends on both model capabilities to reproduce atmospheric dynamics and SS source functions. Aware of the significance of SS representation, further studies could shed light on the results in this direction and future research will focus on an improvement of SS emission. In the new section 6.2.1, we included the next paragraph to summarize what we discussed here: "Large uncertainties persist in modelling sea spray aerosols within climate models Grythe et al. (2014); Lapere et al. (2023), and regional models have shown varying performance of sea salt source functions among different stations (Barthel et al., 2019). Nevertheless, for the group of stations considered in this study, as well as for other locations worldwide (Tegen et al., 2019), the standard SS emission configuration in ECHAM6.3-HAM2.3 provides the most reasonable representation among the available schemes. Although the resulting biases in SS concentrations affect the predicted PMOA values, the evaluation of marine organics discussed in the following sections remains meaningful and valid despite the discrepancies in SS observations."

15. Section 6.2.4. Why there is no scatter plot for comparing total modelled and total measured OM? Is this including anthropogenic sources. I think confusion started at line 269.

We understand the relevance of additionally showing the comparison of total model and measured OM. Therefore, we included a new figure in section 6.2.5 with the modelled PMOA+OC and observed OM (Figure 9). Note that the model tracer OC originates solely from anthropogenic sources or biomass burning, whereas PMOA is from marine origin.

16. Figure 9. Are Pearson coefficients statistically significant? It would best if each bar has P-value attached. It applies to all regression plots too – no regression equation or even a line should be presented unless it is statistically significant.

We fully agree with the reviewer that including the p-value is vital to demonstrate the statistical significance of the correlation between observed and modelled quantities. Unfortunately, we had not incorporated this detail in the region-specific correlation analysis, which, in several cases, yielded non-significant results. However, when all regions are combined, both the correlation and its associated p-value were significant. Nevertheless, given that the Northern Hemisphere is heavily influenced by anthropogenic pollution, biomass burning, and natural fires—diminishing the marine organic aerosol signal—we excluded the northern oceans from our comparison. With this, the correlation increased, and we observed a clear improvement in the more pristine southern regions when PMOA is considered. This result is illustrated by a newly added scatter plot of observed organic aerosol versus modelled OC+PMOA in Figure 10, where the p-value was lower than $2.1 \times 10^{-6}$ for both simulation experiments. This updated analysis required major changes to sections 4.1 and 6.3 which could be found in the revised manuscript version.

**References**

[revised manuscript text omitted]